# LORE: Lagrangian-Optimized Robust Embeddings for Visual Encoders

**Borna Khodabandeh**[1*]    **Amirabbas Afzali**[2*]
**Amirhossein Afsharrad**[1,2]    **Seyed Shahabeddin Mousavi**[1,2]    **Sanjay Lall**[1]
**Sajjad Amini**[3]    **Seyed-Mohsen Moosavi-Dezfooli**[4]

[1]Stanford University    [2]Aktus AI    [3]University of Massachusetts Amherst    [4]Apple

{bornakh, afsharrad, lall}@stanford.edu    ssmousav@cs.stanford.edu
amirabbas.afzali@aktus.ai    samini@umass.edu    smoosavi@apple.com

## Abstract

Visual encoders have become fundamental components in modern computer vision pipelines. However, ensuring robustness against adversarial perturbations remains a critical challenge. Recent efforts have explored both supervised and unsupervised adversarial fine-tuning strategies. We identify two key limitations in these approaches: (i) they often suffer from instability, especially during the early stages of fine-tuning, resulting in suboptimal convergence and degraded performance on clean data, and (ii) they exhibit a suboptimal trade-off between robustness and clean data accuracy, hindering the simultaneous optimization of both objectives. To overcome these challenges, we propose **L**agrangian-**O**ptimized **R**obust **E**mbeddings (LORE), a novel unsupervised adversarial fine-tuning framework. LORE utilizes constrained optimization, which offers a principled approach to balancing competing goals, such as improving robustness while preserving nominal performance. By enforcing embedding-space proximity constraints, LORE effectively maintains clean data performance throughout adversarial fine-tuning. Extensive experiments show that LORE stabilizes training and significantly improves zero-shot adversarial robustness with minimal degradation in clean data accuracy. Furthermore, we demonstrate the effectiveness of the adversarially fine-tuned image encoder in out-of-distribution generalization and enhancing the interpretability of image embeddings. The code is available on GitHub.

## 1   Introduction

In recent years, embeddings from foundation Vision-Language Models (VLMs), such as CLIP [Radford et al., 2021], have transformed downstream tasks, including classification [Conde and Turgutlu, 2021], object detection [Zhong et al., 2021], segmentation [Xu et al., 2022], and image generation [Saharia et al., 2022]. These models, trained on large-scale web data, semantically align inputs from different modalities into a joint embedding space. enabling remarkable zero-shot generalization capabilities like zero-shot image classification, where virtually any class can be encoded via its textual description.[Radford et al., 2021, Qian and Hu, 2024].

However, visual encoders, such as CLIP models, exhibit significant vulnerabilities. Adversarial attacks [Goodfellow et al., 2015] and backdoor vulnerabilities [Bai et al., 2024, Liang et al., 2024] can lead to erroneous and misleading embeddings. These challenges highlight the need for robust alignment methods in large, pre-trained visual encoders. To address this, adversarial fine-tuning plays a crucial role in enhancing the robustness of deep neural networks against adversarial examples

---

*Borna Khodabandeh and Amirabbas Afzali contributed equally to this work.

39th Conference on Neural Information Processing Systems (NeurIPS 2025).

[Pang et al., 2020, Kurakin et al., 2017, Madry et al., 2019]. Moreover, maintaining accuracy and correct embeddings is vital for downstream applications, as it ensures the model's versatility and reliability without compromising clean data performance, i.e., its nominal performance [Tsipras et al., 2019, Dobriban et al., 2022]. While recent works have proposed effective methods to enhance the robustness of visual encoders [Schlarmann et al., 2024, Mao et al., 2023], they often struggle to manage the trade-off between robustness and nominal performance effectively. Therefore, principled approaches are required to explicitly balance this trade-off [Zhang et al., 2019a].

In this work, we focus on the vision encoder and introduce *Lagrangian-Optimized Robust Embeddings* (LORE), a novel unsupervised adversarial fine-tuning framework that effectively balances robustness and nominal performance through a novel constrained optimization framework. While we primarily target CLIP's image encoder, due to its widespread adoption and strong zero-shot capabilities, we also demonstrate the applicability of LORE to other foundation models such as DINOv2 [Oquab et al., 2024]. LORE enforces proximity to a reference model in the embedding space by leveraging the Lagrangian dual method [Bertsekas, 1997], enabling robustness improvements without compromising semantic fidelity on clean data.

Our main contributions are as follows: 1) We propose LORE, a novel unsupervised adversarial fine-tuning framework that achieves a superior empirical balance on the trade-off between robustness and nominal performance, with a hyperparameter ($\rho$) enabling principled control over this trade-off. 2) We demonstrate the effectiveness of LORE in image classification, significantly improving zero-shot adversarial robustness over CLIP while maintaining minimal degradation in clean accuracy. 3) We analyze the behavior of the adversarially fine-tuned LORE-CLIP encoder in terms of out-of-distribution generalization and show that it improves the interpretability of image embeddings, highlighting its advantages over other unsupervised adversarial fine-tuning baselines.

## 2 Related Work

**Adversarial Robustness: Attacks and Defenses.** Adversarial attacks exploit neural network vulnerabilities by crafting imperceptible perturbations, leading to misclassifications [Goodfellow et al., 2015, Szegedy et al., 2014]. Defenses like adversarial training improve robustness but struggle with generalization [Madry et al., 2019]. Balancing robustness and nominal performance remains challenging [Tsipras et al., 2019, Zhang et al., 2019b], necessitating more versatile defenses, especially in multimodal settings. A principled approach to managing the trade-off is Constrained Optimization: [Robey et al., 2021] pioneered using a Lagrangian duality framework to enforce explicit constraints, avoiding heuristic loss weighting.

**Robustness in Foundation Encoders.** Recent studies have highlighted the vulnerability of foundation visual encoders, e.g. CLIP and DINOv2, to adversarial attacks, demonstrating that embeddings can be disrupted [Zhang et al., 2024]. To address this, supervised defenses like TeCoA [Mao et al., 2023] and MMCoA [Zhou et al., 2024] improve robustness by aligning adversarial and clean embeddings, often relying on auxiliary information like text, using simplistic text descriptions, or attention maps,[Yu et al., 2024], assuming normalized embeddings. PMG-AFT [Wang et al., 2024] extends this approach by minimizing the distance between adversarial features and those of the pre-trained model. In the label-free setting, Unsupervised Adversarial Fine-Tuning seeks to achieve robustness by modifying contrastive losses, leading to methods like Robust Contrastive Learning (RoCL) [Kim et al., 2020] and the decoupled approach of DeACL [Zhang et al., 2022]. Among unsupervised fine-tuning strategies for the CLIP vision encoder, FARE [Schlarmann et al., 2024] stands out as a direct baseline. However, FARE and similar methods often lack explicit mechanisms to balance robustness and task performance, resulting in a suboptimal trade-off. Due to its unsupervised nature, we consider FARE a baseline for evaluating our proposed approach.

## 3 Background

**Adversarial Attacks.** Adversarial attacks aim to deceive machine learning models by adding imperceptible perturbations to inputs. A prominent example is the Projected Gradient Descent (PGD) attack, which iteratively maximizes a loss function $J$ (e.g., cross-entropy) to craft adversarial examples. Starting from an input $x$, PGD updates:

$$x'_{t+1} = \text{Proj}_{\mathcal{B}(x,\varepsilon)} \left( x'_t + \eta \cdot \text{sign}(\nabla_{x'_t} J(x'_t, y)) \right), \tag{1}$$

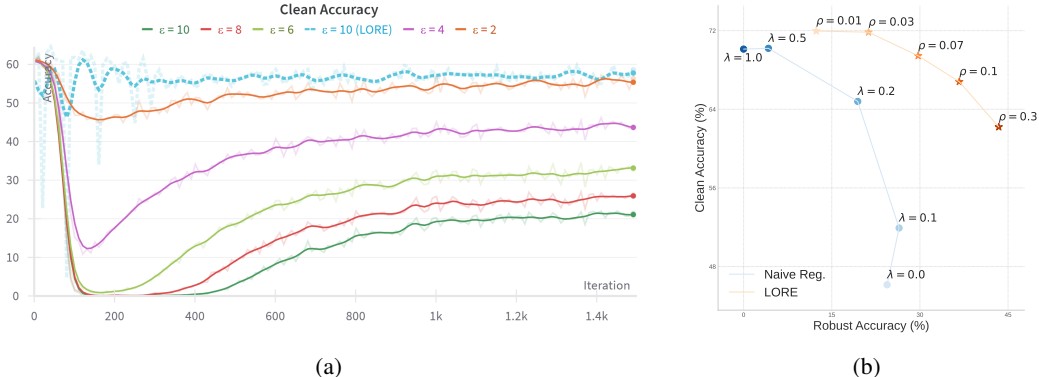

(a)              (b)

Figure 1: **(a)** clean data accuracy during adversarial fine-tuning with different training perturbation strengths $\varepsilon$, using the loss from Eq. (2). Larger $\varepsilon$ values result in substantial drops in clean data accuracy and early training instability. LORE (with $\rho = 0.1$) mitigates this effect, even on ($\varepsilon = 10$), maintaining stable and higher clean data accuracy. **(b)** Pareto frontier comparison between naive regularization of Eq. (2) (blue, varying $\lambda$) and LORE (orange, varying $\rho$). LORE yields a strictly better empirical Pareto front, demonstrating superior trade-offs.

where $y$ is the true label, $\eta$ is the step size, and $\varepsilon$ bounds the $\ell_\infty$-norm of the perturbation. The projection $\text{Proj}_{\mathcal{B}(x,\varepsilon)}$ ensures $x'$ stays within the allowed perturbation set. This procedure reveals the susceptibility of models like CLIP to adversarial manipulations.

**Unsupervised Adversarial Fine-Tuning.** FARE [Schlarmann et al., 2024] proposed an unsupervised method for fine-tuning CLIP vision encoder $\phi$. FARE's goal is to minimize the Euclidean distance between clean and perturbed embeddings under worst-case perturbation conditions:

$$\mathcal{L}_{\text{FARE}}(\phi_\theta, x) = \max_{\delta:\|\delta\|_\infty \leq \varepsilon} \|\phi_\theta(x + \delta) - \phi_{\theta_0}(x)\|_2^2, \tag{2}$$

where $\phi_{\theta_0}$ refers to the original (frozen) CLIP encoder, and $\delta$ represents the adversarial perturbation. Unlike methods like TeCOA and MMCoA, FARE does not require text inputs or labels, relying solely on the image encoder. This loss function aims to (i) enhance the robustness of image embeddings generated by $\phi_\theta$, and (ii) ensure that the image encoder $\phi_\theta$ remains close to the original encoder $\phi_{\theta_0}$ to maintain performance on clean data, preserving nominal performance.

## 4 Preliminary Experiments

To motivate our approach, we conduct a series of preliminary experiments that reveal critical limitations in adversarial fine-tuning with the loss defined in Eq. (2). In particular, we examine two key challenges: (i) instability during early fine-tuning, and (ii) poor robustness-accuracy trade-offs under naive regularization.

**(i)** Fig. 1a depicts the clean data accuracy of a CLIP image encoder during adversarial fine-tuning using the loss function defined in (2). The results indicate substantial declines in clean data accuracy, particularly at elevated perturbation strengths (e.g., $\varepsilon = 8, 10$)[2] . Additionally, there is notable instability during the early stages of training. This instability hampers optimal convergence and diminishes performance on clean data. Consequently, the model may converge to local minima, losing significant pretrained knowledge, despite achieving robustness.

**(ii)** As a straightforward approach to mitigate the degradation of clean data accuracy, one may introduce a naive regularization term to the loss (2), aiming to minimize the following objective:

$$\mathcal{L}_{\text{FARE}}(\phi, x) + \lambda \|\phi_\theta(x) - \phi_{\text{org}}(x)\|_2^2, \tag{3}$$

where $\lambda$ serves as a hyperparameter to regulate the strength of the regularization. We assessed the trade-off between clean and robust accuracy by plotting the Pareto frontier across various regularization strengths, as illustrated in Fig. 1b. The blue curve represents naive regularization (with

---

[2]Throughout the paper, all values of $\varepsilon$ refer to normalized $\ell_\infty$ perturbation bounds, i.e., $\varepsilon/255$. For example, $\varepsilon = 2$ corresponds to a perturbation magnitude of $2/255$.

**Algorithm 1** Lagrangian-Optimized Robust Embeddings (LORE)

---

**for** each epoch **do**
    **for** batch $x \sim \mathcal{D}$ **do**
        $\delta \leftarrow \text{ATTACKALG}\big(d(\phi_{\theta_t}(x+\delta), \phi_{\theta_0}(x))\big)$
        **for** $i = 1$ to $K$ **do**
            $\mathcal{L}_{\text{robust}} \leftarrow d(\phi_{\theta_t}(x+\delta), \phi_{\theta_0}(x))$
            $\mathcal{L}_{\text{clean}} \leftarrow d(\phi_{\theta_t}(x), \phi_{\theta_0}(x)) - \rho\, m(x)$
            $\mathcal{L}_{\text{total}} \leftarrow \mathcal{L}_{\text{robust}} + \lambda_\omega(x) \cdot \mathcal{L}_{\text{clean}}$
            $\theta \leftarrow \theta - \eta_\theta \cdot \nabla_\theta \mathcal{L}_{\text{total}}$
        **end for**
        $\omega \leftarrow \omega + \eta_\omega \cdot \mathcal{L}_{\text{clean}} \cdot \nabla_\omega \lambda_\omega(x)$
    **end for**
**end for**
**return** $\theta, \omega$

---

Figure 2: (**Left**) Detailed steps of LORE. (**Right**) A conceptual overview of LORE. Given a clean input, the pre-trained model $\phi_{\theta_0}(x)$ serves as an anchor. LORE optimizes the encoder $\phi_{\theta_t}(x)$ to maintain proximity to $\phi_{\theta_0}(x)$ for clean inputs, while also improving robustness against adversarial perturbations $\phi_{\theta_t}(x+\delta)$, which are pushed away from the anchor. The green region denotes the constraint margin, preserving semantic alignment and mitigating degradation in nominal performance.

varying $\lambda$), revealing a steep trade-off, with robustness gains costing a considerable expense to clean data accuracy. Conversely, the proposed method, LORE (depicted by orange stars, with varying $\rho$), demonstrates a superior Pareto front, consistently attaining higher robust accuracy at comparable or enhanced levels of clean data accuracy. Moreover, LORE maintains stable clean data accuracy throughout training (Fig. 1a), even with large perturbation strengths.

## 5 LORE: Lagrangian-Optimized Robust Embeddings

Our findings in Section 4 highlight the need for a principled method that both stabilizes adversarial fine-tuning and offers a favorable robustness-accuracy trade-off. Constrained learning offers a structured approach to balancing competing objectives like robustness and nominal accuracy. The constrained optimization problem for adversarial robustness in classification, as explored in Robey et al. [2021], can be formulated as:

$$\min_{\theta \in \Theta} \mathbb{E}_{(x,y)\sim\mathcal{D}} \left[ \max_{\delta \in \Delta} \ell(f_\theta(x+\delta), y) \right],$$
$$\text{s.t.} \quad \ell(f_\theta(x), y) \leq \rho, \quad \text{for almost every } (x,y) \in \mathcal{D}, \tag{4}$$

where $\ell$ is the loss function, $\rho \geq 0$ controls the nominal performance level, $f_\theta$ is the classifier parameterized by $\theta$, and $\Delta$ denotes the set of possible perturbations. The presence of infinitely many constraints indexed by the input space makes this a semi-infinite learning problem, which allows the application of tools from semi-infinite optimization theory.

Extending the optimization framework of (4) to an unsupervised setting, we propose enforcing proximity to a reference model $\phi_{\theta_0}$ (e.g., a pre-trained model) in the embedding space $\mathbb{R}^k$, where $k$ denotes the embedding dimension, thereby eliminating the dependence on $y$. Our formulation is:

$$\min_{\theta \in \Theta} \mathbb{E}_{x\sim\mathcal{D}} \left[ \max_{\delta \in \Delta} d(\phi_\theta(x+\delta), \phi_{\theta_0}(x)) \right],$$
$$\text{s.t.} \quad d(\phi_\theta(x), \phi_{\theta_0}(x)) \leq \rho\, m(x), \quad \text{for almost every } x \in \mathcal{D}. \tag{5}$$

Here, $d : \mathbb{R}^k \times \mathbb{R}^k \to \mathbb{R}^+$ is a divergence metric, and $m : \mathcal{X} \to \mathbb{R}^+$ is a sample-specific tolerance margin, independent of $\theta$, used to modulate the constraint across inputs, defining a per-sample margin within which deviations are considered acceptable. For example, when using Euclidean distance, setting $m(x) = \|\phi_{\theta_0}(x)\|_2$ ensures the constraint becomes scale-invariant with respect to the original embedding magnitude.

**Solving the Constrained Problem.** Constrained optimization in deep learning presents significant challenges due to the high-dimensional and non-convex nature of neural networks. Therefore,

following [Robey et al., 2021, Chamon et al., 2020, Chamon and Ribeiro, 2021], we leverage Lagrangian duality to obtain approximate solutions for (5) that generalize effectively with respect to both adversarial and nominal performance. Specifically, we solve the dual optimization problem:

$$\max_{\omega \in \Omega} \min_{\theta \in \Theta} \mathbb{E}_{x \sim \mathcal{D}} \left[ \max_{\delta \in \Delta} d\left(\phi_\theta(x + \delta), \phi_{\theta_0}(x)\right) + \lambda_\omega(x)\left(d\left(\phi_\theta(x), \phi_{\theta_0}(x)\right) - \rho\, m(x)\right) \right], \quad (6)$$

where $\lambda_\omega : \mathcal{X} \to \mathbb{R}^+$ is the dual network parameterized by $\omega$, with outputs made non-negative via Softplus activation, which ensures valid Lagrange multipliers and avoids the sparse gradients of ReLU. This network simplifies the original optimization over all functions $\lambda : \mathcal{X} \to \mathbb{R}^+$. By jointly optimizing $\theta$ and $\omega$, our method achieves two key objectives: (i) enhancing the robustness of the model's embeddings against adversarial examples through the worst-case perturbation $\delta$; and (ii) maintaining proximity to the reference model's embeddings on clean data, i.e., $\phi_{\theta_0}(x)$, to preserve nominal performance. Note that this adaptive framework fundamentally differs from regularization, and is automatically enforced if embeddings diverge, The theoretical gaps and limitations introduced by our formulation and parameterization are detailed in Appendix B.

**Practical Implementation.** For computational efficiency, we implement the dual network as $\lambda_\omega(x) = F_\omega(\phi_{\theta_0}(x))$, where $F_\omega$ is a lightweight two-layer MLP that takes the reference model's embeddings as input. This leverages readily available semantic information from $\phi_{\theta_0}(x)$ for input-dependent constraint weighting with no extra embedding computation. In Appendix G, we provide a detailed analysis of the $\lambda_\omega$ architecture and its impact on performance. Specifically, we empirically demonstrate that modeling the dual variable as an input-dependent network, i.e., $\lambda_\omega(x)$, significantly outperforms the input-independent variant $\lambda_\omega$. We instantiate the divergence metric as the squared $\ell_2$ distance, $d(z_1, z_2) = \|z_1 - z_2\|_2^2$, and set $m(x) = \|\phi_{\theta_0}(x)\|_2^2$ to enforce scale-invariant constraints. Substituting these choices into our general framework yields the following optimization problem:

$$\max_{\omega \in \Omega} \min_{\theta \in \Theta} \mathbb{E}_{x \sim \mathcal{D}} \left[ \max_{\delta \in \Delta} \|\phi_\theta(x + \delta) - \phi_{\theta_0}(x)\|_2^2 + \lambda_\omega(x)\left(\|\phi_\theta(x) - \phi_{\theta_0}(x)\|_2^2 - \rho\|\phi_{\theta_0}(x)\|_2^2\right) \right]. \quad (7)$$

The adversarial perturbation $\delta$ is approximated using projected gradient descent (PGD) within an $\ell_\infty$-bounded set. Problem (7) employs a *primal-dual* framework to balance robustness and accuracy dynamically. The *primal* step optimize $\phi_\theta$ for robustness while maintaining proximity to $\phi_{\theta_0}$ on clean data. The *dual* step updates parameters $\omega$ to strengthen constraints where $d(\phi_\theta(x), \phi_{\theta_0}(x))$ exceeds $\rho\, m(x)$, preventing overly conservative solutions. This approach is crucial for vision-language models, ensuring semantic alignment via embedding proximity, and eliminates heuristic loss weighting by treating robustness and accuracy as competing objectives within a Lagrangian duality framework.

Intuitively, our scale-invariant constraint $\|\phi_\theta(x) - \phi_{\theta_0}(x)\|_2^2 \leq \rho\|\phi_{\theta_0}(x)\|_2^2$ preserves angular relationships. As detailed in Appendix F, this bounds the deviation in cosine similarity $S_C(u, v) = u^T v/(\|u\|\|v\|)$, ensuring $|S_C(\phi_{\theta_0}(x), v) - S_C(\phi_\theta(x), v)| \leq 2\sqrt{\rho}$ for any vector $v \in \mathbb{R}^n$. This preserves image-text alignment in vision-language models during adversarial fine-tuning.

**Optimization.** We optimize the LORE objective using an alternating training procedure. For each batch $x$ sampled from the dataset $\mathcal{D}$, we first generate adversarial examples $\delta$. Then, we perform $K$ steps of gradient descent to update the encoder parameters $\theta$. After the $K$ primal updates, we take a single gradient step to update the dual network parameters $\omega$, using the clean loss $\mathcal{L}_{\text{clean}}$ (as defined in Algorithm 1) to guide the adjustment of $\lambda_\omega(x)$. This alternating update strategy enables the model to dynamically balance robustness and clean performance, with $K$ acting as a hyperparameter that controls the frequency of primal updates. The full procedure is summarized in Algorithm 1. Although non-convex convergence guarantees are challenging, this alternating optimization is empirically stable and effective in our experiments. Ablation studies on the choice of $K$ and constraint threshold $\rho$ are provided in Appendix I.

## 6 Main Results

Although LORE and FARE optimize the identical adversarial loss, LORE's proximity constraints serve as a non-trivial, sample-based regularizer that steers training away from poor local minima, yielding consistently stronger robustness without sacrificing clean accuracy. We now present a suite of experiments to substantiate this claim.

We begin by demonstrating how LORE enables controlled trade-offs between robustness and nominal performance. Next, we evaluate zero-shot image classification under both clean and adversarial

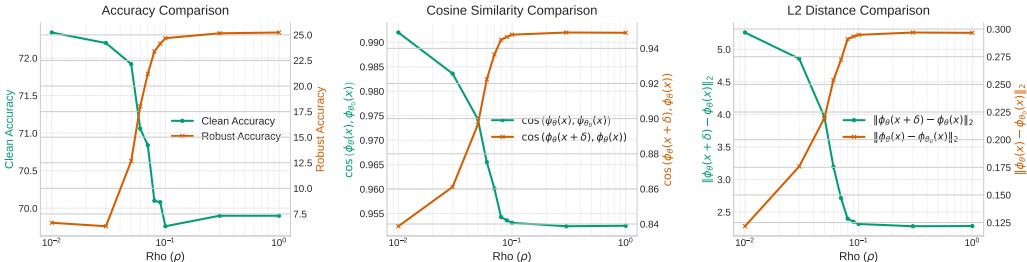

Figure 3: Influence of constraint threshold $\rho$ on model behavior. As $\rho$ increases, robustness improves at the cost of clean data accuracy, cosine alignment, and embedding fidelity, highlighting the effectiveness of controlling the trade-off between robustness and fidelity by tuning $\rho$ in LORE.

settings. We then analyze LORE's benefits for embedding interpretability and out-of-distribution robustness. Further results and ablations are provided in Appendix J.

**Experimental Setup.** To empirically validate our approach, we conduct the majority of our experiments using the ViT-B/32 CLIP model. In the second part of our evaluation (Section 6.3), we further assess the effectiveness of LORE on alternative architectures such as ConvNeXt-B CLIP and DINOv2. We use both ImageNet and ImageNet-100, a curated subset of ImageNet, as training datasets throughout our experiments. Additional details for each experiment, including figures and tables, are provided in Appendix C. Throughout the results, superscripts denote the $\ell_\infty$ perturbation bound $\varepsilon$ used during adversarial fine-tuning; for instance, LORE[2] refers to a model trained with $\varepsilon = 2/255$. We consistently compare our method with FARE, which serves as the unconstrained counterpart to LORE and also utilizes unsupervised adversarial fine-tuning. All models were trained for 2 epochs on ImageNet or 5 epochs on ImageNet-100. Experiments were conducted using 8 NVIDIA HGX H100 80GB GPUs.

## 6.1 Controlling the Robustness-Accuracy Trade-off

The parameter $\rho$ directly governs the allowed clean embedding deviation, offering a more precise control knob for the robustness-accuracy trade-off than heuristic weightings. While such trade-offs have been extensively studied in supervised adversarial robustness [Zhang et al., 2019a, Xiao et al., 2024, Javanmard et al., 2020], they remain underexplored in unsupervised adversarial fine-tuning.

We apply our constrained optimization framework to probe robustness-accuracy trade-offs. Using a proximity constraint, we restrict the hypothesis space from the full family of encoders $\mathcal{H}$ to the fidelity-constrained subset $\mathcal{H}_\rho = \left\{ \phi \in \mathcal{H} \mid d(\phi(x), \phi_{\theta_0}(x)) \leq \rho \, m(x) \text{ for a.e. } x \in \mathcal{X} \right\}$. Intuitively, $\mathcal{H}_\rho$ is a neighborhood of the clean encoder $\phi_{\theta_0}$: for small $\rho$, it's a tight ball around $\phi_{\theta_0}$, and as $\rho$ grows it expands until it recovers the full class $\mathcal{H}$.

We then define the pointwise adversarial loss as $\ell_{\mathrm{adv}}(\phi, x; \phi_{\theta_0}) \triangleq \max_{\delta \in \Delta} \, d\big(\phi(x + \delta), \phi_{\theta_0}(x)\big)$, which induces the optimization

$$\phi_\rho^* = \underset{\phi \in \mathcal{H}_\rho}{\arg\min} \, \mathbb{E}_{x \sim \mathcal{D}}\big[\ell_{\mathrm{adv}}(\phi, x; \phi_{\theta_0})\big] \triangleq \underset{\phi \in \mathcal{H}_\rho}{\arg\min} \, \ell_{\mathrm{adv}}(\phi; \phi_{\theta_0}). \tag{8}$$

Constraining the search space stabilizes training by avoiding extreme encoder shifts. However, it does so at the cost of theoretical optimality—since restricting $\mathcal{H}$ may exclude the global minimizer. In practice, the constrained formulation can nonetheless outperform the unconstrained one by steering clear of poor local minima.

**Assumption 6.1.** Assume $d(\cdot, \cdot)$ is a nonnegative metric on the embedding space. Moreover, for any fixed reference embedding $c$, the map $u \mapsto d(u, c)$ is $L_d$-Lipschitz in its first argument, i.e., $\big|d(u_1, c) - d(u_2, c)\big| \leq L_d \, \|u_1 - u_2\|$ for some norm $\|\cdot\|$.

**Theorem 6.2** (Robustness Suboptimality Bounds). *Let:*

$$R = \min_{\phi \in \mathcal{H}} \ell_{\mathrm{adv}}(\phi; \phi_{\theta_0}), \qquad R_\rho = \min_{\phi \in \mathcal{H}_\rho} \ell_{\mathrm{adv}}(\phi; \phi_{\theta_0}).$$

*Since $\mathcal{H}_\rho \subset \mathcal{H}$, we have $R_\rho \geq R$. Moreover, the suboptimality gap satisfies:*

$$0 \leq R_\rho - R \leq \sqrt{k}\left(L_\rho^* + L'\right)\varepsilon + \|\phi_\rho^* - \phi^*\|.$$

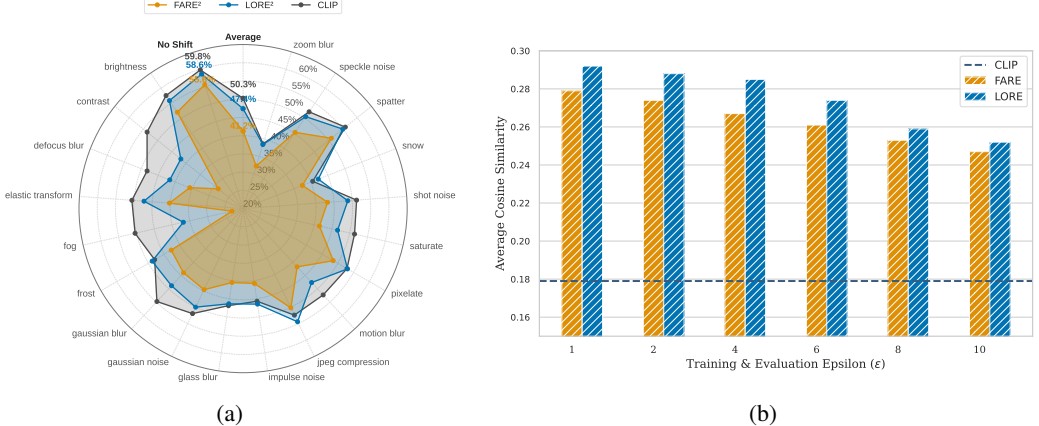

(a)                                                              (b)

Figure 4: **(a)** Robustness to common corruptions on ImageNet-C as an OOD evaluation. **(b)** Embedding interpretability assessment based on average cosine similarity between clean image embeddings and 150 corresponding GPT4-generated text templates.

*Here $\phi^*$ and $\phi_\rho^*$ are the minimizers in $\mathcal{H}$ and $\mathcal{H}_\rho$, respectively; $L_\rho^*$ and $L'$ are their associated Lipschitz constants; and adversarial perturbations are $\ell_\infty$-bounded by $\varepsilon$ in $\mathbb{R}^k$. (Proof in Appendix A.)*

The bound shows that the excess loss $R_\rho - R$ grows with both the attack radius $\varepsilon$ and the encoder shift $\|\phi_\rho^* - \phi^*\|$. While we employ squared $\ell_2$ geometry for its optimization convenience, this choice only affects the numerical constants—absorbed into the embedding norms and local Lipschitz terms.

Intuitively, increasing $\rho$ relaxes the proximity constraint, granting the model more flexibility (and thus greater robustness) at the expense of nominal accuracy, whereas smaller $\rho$ keeps the model tightly aligned with the pre-trained encoder, favoring clean performance.

We empirically evaluate LORE's control over the robustness–accuracy frontier by training ten models with varying $\rho$. Figure 1b shows the resulting Pareto frontier compared to additive-regularization baselines (Eq. 2), and Figure 3 reports clean-data accuracy and adversarial success rate on the test set. These experiments confirm both the practical tightness of our theoretical bound and the superior trade-off achieved by our constrained-optimization framework.

## 6.2 Out-of-Distribution Robustness and Embedding Interpretability

To assess how unsupervised adversarial fine-tuning affects out-of-distribution (OOD) robustness, we evaluate a ViT-B/32 CLIP model fine-tuned with LORE on ImageNet, using the ImageNet-C [Hendrycks and Dietterich, 2019], which applies common corruptions to simulate distribution shifts. Since adversarial fine-tuning often reduces clean accuracy, it can lead to degraded OOD performance. As shown in Fig. 4a, FARE[2] suffers a substantial drop under corruptions, whereas LORE[2] maintains much stronger generalization. In fact, for certain perturbations (e.g., *jpeg compression*), LORE[2] even outperforms the original pretrained CLIP model, underscoring its robustness to distribution shift.

Adversarial training is also known to enhance embedding interpretability and cross-modal alignment [Croce et al., 2025]. To measure this effect in the CLIP vision–language space, we generate 150 natural-language templates using GPT-4 (e.g., "This is a photo of { }") to capture class semantics, then compute the average cosine similarity between each clean image embedding and its corresponding text embedding. Figure 4b shows that both FARE[2] and LORE[2] improve over the pretrained baseline, but LORE[2] consistently achieves higher alignment across all training and evaluation $\varepsilon$ settings, demonstrating superior semantic representation.

## 6.3 Image Classification

**Zero-shot Image Classification.** We evaluate adversarial robustness of the ViT-B/32 CLIP vision encoder on 13 zero-shot benchmarks from `CLIP-benchmark`[3], all originally trained on ImageNet. For each dataset, we compare LORE against FARE under both clean and adversarial conditions.

---

[3]https://github.com/LAION-AI/CLIP_benchmark

Table 1: Clean and adversarial accuracy for in-domain image classification on ImageNet-100 across different CLIP vision encoders, evaluated using the APGD attack.

| Method | Backbone | Clean | $\varepsilon=1$ | $\varepsilon=2$ | $\varepsilon=4$ | $\varepsilon=8$ |
|---|---|---|---|---|---|---|
| FARE[2] | ViT-B/16 | 70.40 | 53.0 | 34.9 | 8.8 | 0.06 |
| LORE[2] | ViT-B/16 | **74.7** | 62.3 | 47.7 | 20.8 | 0.74 |
| FARE[4] | ViT-B/16 | 58.1 | 47.7 | 37.1 | 19.0 | 2.22 |
| LORE[4] | ViT-B/16 | 71.5 | **62.3** | **53.3** | **34.7** | **9.06** |
| FARE[2] | ViT-B/32 LAION | 65.4 | 41.0 | 19.0 | 2.02 | 0.02 |
| LORE[2] | ViT-B/32 LAION | **70.2** | **51.8** | **31.4** | 7.26 | 0.04 |
| FARE[4] | ViT-B/32 LAION | 52.7 | 36.7 | 23.4 | 6.72 | 0.20 |
| LORE[4] | ViT-B/32 LAION | 68.4 | 44.7 | 29.6 | 10.7 | 0.62 |
| FARE[2] | ConvNeXt-B | 74.2 | 61.6 | 46.1 | 16.7 | 0.22 |
| LORE[2] | ConvNeXt-B | **75.6** | 64.9 | 52.4 | 25.6 | 1.04 |
| FARE[4] | ConvNeXt-B | 70.6 | 61.6 | 52.3 | 32.7 | 6.48 |
| LORE[4] | ConvNeXt-B | 73.5 | **66.0** | **58.1** | **40.3** | **10.4** |

Table 2: Clean and adversarial accuracy for in-domain image classification on ImageNet across different DINOv2 variants. Adversarial robustness is evaluated using APGD attack.

| Method | Backbone | Clean | $\varepsilon=1$ | $\varepsilon=2$ | $\varepsilon=4$ | $\varepsilon=8$ |
|---|---|---|---|---|---|---|
| FARE[4] | ViT-S/14 | 69.2 | 60.7 | **51.2** | 30.7 | 2.91 |
| LORE[4] | ViT-S/14 | **77.3** | **60.8** | 50.0 | 30.3 | 5.8 |
| FARE[8] | ViT-S/14 | 55.1 | 48.9 | 42.7 | 30.0 | 8.13 |
| LORE[8] | ViT-S/14 | 75.1 | 55.9 | 48.8 | **36.8** | **13.7** |
| FARE[4] | ViT-B/14 | 78.3 | 71.9 | 64.1 | 44.0 | 6.51 |
| LORE[4] | ViT-B/14 | 80.2 | **73.5** | **67.1** | **49.6** | 11.2 |
| FARE[8] | ViT-B/14 | 69.4 | 63.8 | 57.8 | 44.1 | 16.0 |
| LORE[8] | ViT-B/14 | **80.5** | 65.0 | 59.7 | 48.5 | **21.8** |

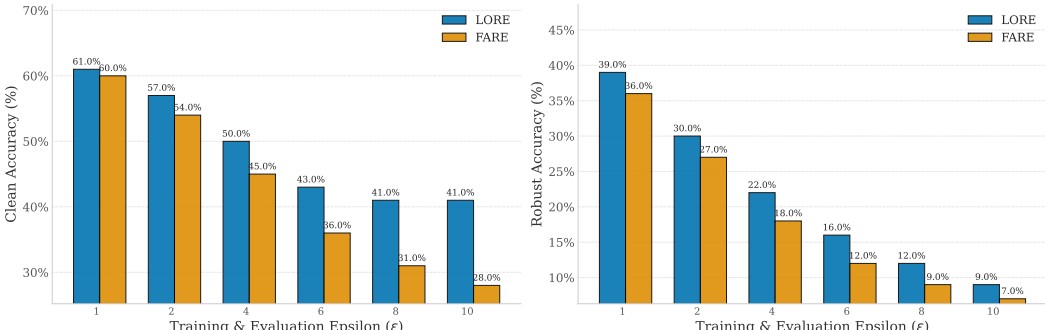

Figure 5: Comparison of LORE and FARE across different training and evaluation perturbations ($\varepsilon$). LORE consistently outperforms FARE, particularly at higher $\varepsilon$ values, achieving higher robust accuracy while maintaining better clean performance, especially at higher perturbation intensities.

Robustness is measured using the first two AutoAttack methods [Croce and Hein, 2020]: APGD with cross-entropy loss and APGD with targeted DLR loss, each run for 100 iterations under an $\ell_\infty$ constraint. As shown in Table 11, the base CLIP model has negligible adversarial robustness, whereas LORE consistently outperforms FARE across all settings, improving adversarial accuracy without sacrificing clean performance.

**In-domain Image Classification.** We further assess LORE on in-domain tasks across various vision architectures. Table 1 reports clean and adversarial accuracy on ImageNet-100 for several CLIP-style backbones using APGD. We also fine-tune the DINOv2 visual encoder with a fixed classification head on the full ImageNet dataset. Results in Table 2 show that LORE enhances robustness not only for CLIP-style models but also for other foundation models such as DINOv2 [Oquab et al., 2024], demonstrating its broad applicability across visual encoder architectures[4].

**Robustness at High Adversarial Intensity.** A key challenge in improving robustness against large adversarial perturbations is the instability introduced by high training perturbation budgets ($\varepsilon$). Training with large $\varepsilon$ values often causes a sharp drop in nominal performance and a loss of alignment with the original model's semantics, making naive adversarial fine-tuning impractical in high-$\varepsilon$ regimes. As illustrated in Fig. 1a and further supported by higher evaluation $\varepsilon$ in Fig. 5, this degradation is particularly evident in the early stages of training. Appendix D provides further insight, showing that adversarial fine-tuning using the loss in Eq. (2) rapidly disrupts the proximity between clean inputs and their reference embeddings. This degradation intensifies with increasing $\varepsilon$, resulting in a more pronounced collapse in clean data accuracy. In contrast, Fig. 6 compares LORE and FARE, showing that LORE explicitly enforces constraint satisfaction over time, thereby preserving proximity between clean embeddings and their pre-trained references, while FARE exhibits a progressive increase in deviation during training. This constraint-guided optimization plays a critical role in mitigating early instability and enables LORE to achieve robust and stable learning even under high adversarial intensity.

---

[4]For each model, bold font highlights the best result, and underlined text denotes the second-best result.

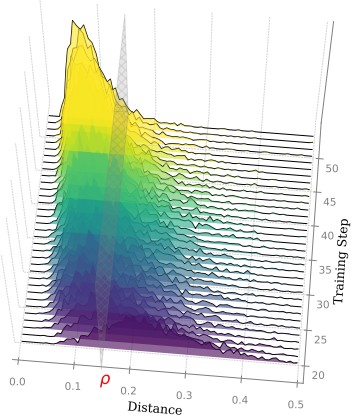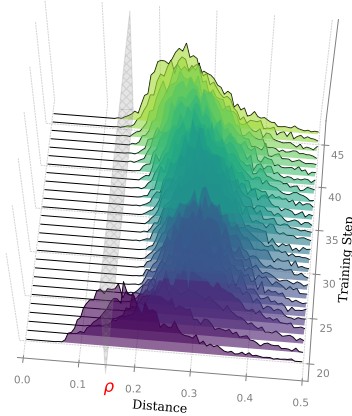

Figure 6: The plot shows the $\ell_2$ distance between the fine-tuned clean embedding, $\phi_\theta(x)$, and the frozen reference embedding, $\phi_{\theta_0}(x)$, over training steps. **(Left)** LORE[4] effectively regulates this deviation through its constraint-aware mechanism, gradually realigning the clean embeddings within the $\rho$ threshold. **(Right)** FARE[4], which lacks such a constraint, exhibits a significant upward drift in the $\ell_2$ distance, indicating a collapse in nominal performance and embedding fidelity.

Table 3: A comprehensive evaluation of clean and adversarial performance is conducted across various image classification datasets using the ViT-B/32 CLIP model. All models are trained on ImageNet and evaluated in a zero-shot setting across diverse benchmarks. Our method consistently achieves a performance increase (↑) relative to the corresponding FARE models.

| Eval. | Vision encoder | ImageNet | CalTech | Cars | CIFAR10 | CIFAR100 | DTD | EuroSAT | FGVC | Flowers | ImageNet-R | ImageNet-S | PCAM | OxfordPets | STL-10 | Average Zero-shot | |
|---|---|---|---|---|---|---|---|---|---|---|---|---|---|---|---|---|---|
| | | | | | | | | Zero-shot datasets | | | | | | | | | |
| clean | CLIP | 59.8 | 84.1 | 59.6 | 89.7 | 63.3 | 44.4 | 46.1 | 19.6 | 66.3 | 69.3 | 42.3 | 62.3 | 87.5 | 97.2 | 64.0 | |
| | FARE[1] | 56.6 | 84.0 | 56.3 | 86.4 | 61.1 | 40.5 | 27.2 | 18.1 | 62.0 | 66.4 | 40.5 | 55.5 | 86.1 | 95.8 | 60.0 | |
| | LORE[1] | 57.4 | 84.4 | 55.9 | 88.5 | 64.5 | 40.1 | 29.9 | 16.7 | 61.3 | 67.2 | 41.5 | 53.8 | 86.9 | 96.3 | 60.5 | ↑0.5 |
| | FARE[2] | 52.9 | 82.2 | 49.7 | 76.3 | 51.1 | 36.4 | 18.4 | 15.7 | 53.3 | 60.4 | 35.9 | 48.2 | 82.7 | 93.0 | 54.1 | |
| | LORE[2] | 55.7 | 83.0 | 51.0 | 83.4 | 59.7 | 37.2 | 23.0 | 15.9 | 54.5 | 63.4 | 39.3 | 51.2 | 84.3 | 94.5 | 57.0 | ↑2.9 |
| | FARE[4] | 42.6 | 78.1 | 36.5 | 55.9 | 35.8 | 28.8 | 15.7 | 10.6 | 36.1 | 49.3 | 27.1 | 50.0 | 71.8 | 85.6 | 44.7 | |
| | LORE[4] | 50.1 | 80.3 | 40.1 | 72.4 | 49.6 | 32.4 | 17.7 | 11.4 | 39.7 | 55.1 | 33.6 | 50.0 | 79.3 | 90.4 | 50.2 | ↑5.5 |
| $\varepsilon = 1.0$ | CLIP | 0.0 | 0.0 | 0.0 | 0.0 | 0.0 | 0.0 | 0.0 | 0.0 | 0.0 | 0.0 | 0.1 | 0.0 | 0.0 | 0.0 | 0.0 | |
| | FARE[1] | 27.8 | 68.6 | 16.1 | 61.0 | 35.6 | 22.5 | 6.1 | 2.9 | 30.6 | 34.4 | 22.5 | 24.7 | 55.8 | 82.2 | 35.6 | |
| | LORE[1] | 32.9 | 71.0 | 18.7 | 67.1 | 40.0 | 23.7 | 9.4 | 4.2 | 33.5 | 37.6 | 24.8 | 28.3 | 60.5 | 84.1 | 38.7 | ↑3.1 |
| | FARE[2] | 34.3 | 75.2 | 22.6 | 60.1 | 35.4 | 24.7 | 12.6 | 5.3 | 33.9 | 39.7 | 24.1 | 30.4 | 64.8 | 83.3 | 39.4 | |
| | LORE[2] | 39.3 | 76.3 | 23.3 | 67.0 | 43.2 | 26.4 | 12.3 | 6.5 | 35.8 | 42.4 | 26.4 | 39.0 | 68.5 | 85.6 | 42.5 | ↑3.1 |
| | FARE[4] | 33.2 | 74.8 | 21.4 | 44.9 | 28.0 | 22.4 | 14.0 | 5.8 | 27.3 | 37.1 | 21.3 | 50.2 | 59.3 | 77.7 | 37.2 | |
| | LORE[4] | 41.8 | 77.2 | 24.1 | 61.2 | 39.9 | 24.5 | 14.3 | 7.8 | 30.2 | 41.6 | 25.5 | 50.2 | 68.8 | 83.2 | 42.2 | ↑5.0 |
| $\varepsilon = 2.0$ | CLIP | 0.0 | 0.0 | 0.0 | 0.0 | 0.0 | 0.0 | 0.0 | 0.0 | 0.0 | 0.0 | 0.1 | 0.0 | 0.0 | 0.0 | 0.0 | |
| | FARE[1] | 8.0 | 43.5 | 1.9 | 31.0 | 14.7 | 12.9 | 0.6 | 0.2 | 6.8 | 13.4 | 11.7 | 14.1 | 15.9 | 54.9 | 17.0 | |
| | LORE[1] | 13.1 | 49.0 | 3.3 | 37.9 | 19.0 | 14.2 | 2.5 | 0.5 | 10.1 | 17.6 | 13.1 | 19.1 | 23.1 | 61.2 | 20.8 | ↑3.8 |
| | FARE[2] | 19.3 | 59.9 | 7.7 | 41.2 | 22.8 | 17.8 | 9.6 | 1.5 | 16.4 | 24.2 | 15.9 | 23.4 | 38.6 | 68.6 | 26.7 | |
| | LORE[2] | 24.0 | 63.3 | 8.6 | 47.2 | 27.2 | 18.2 | 10.6 | 1.7 | 18.5 | 26.0 | 18.4 | 28.0 | 44.4 | 73.1 | 29.6 | ↑2.9 |
| | FARE[4] | 24.1 | 65.5 | 10.4 | 36.0 | 21.6 | 18.8 | 12.3 | 2.7 | 17.9 | 27.7 | 15.8 | 50.0 | 44.4 | 68.8 | 30.1 | |
| | LORE[4] | 32.6 | 69.5 | 12.4 | 50.8 | 29.6 | 20.9 | 13.0 | 3.3 | 21.6 | 32.3 | 20.0 | 50.1 | 55.9 | 76.1 | 35.0 | ↑4.9 |
| $\varepsilon = 4.0$ | CLIP | 0.0 | 0.0 | 0.0 | 0.0 | 0.0 | 0.0 | 0.0 | 0.0 | 0.0 | 0.0 | 0.1 | 0.0 | 0.0 | 0.0 | 0.0 | |
| | FARE[1] | 0.3 | 6.3 | 0.0 | 1.7 | 2.0 | 2.3 | 0.0 | 0.0 | 0.1 | 2.6 | 2.4 | 0.9 | 0.0 | 5.3 | 1.8 | |
| | LORE[1] | 0.7 | 9.7 | 0.0 | 3.5 | 3.1 | 4.0 | 0.0 | 0.0 | 0.2 | 3.8 | 2.8 | 2.7 | 0.0 | 9.4 | 3.0 | ↑1.2 |
| | FARE[2] | 3.2 | 27.5 | 0.5 | 12.3 | 7.0 | 7.7 | 4.3 | 0.0 | 2.4 | 6.8 | 5.1 | 15.8 | 3.0 | 30.1 | 9.4 | |
| | LORE[2] | 5.7 | 31.1 | 0.7 | 13.0 | 8.2 | 9.7 | 0.8 | 0.0 | 3.1 | 8.3 | 6.5 | 18.2 | 7.2 | 33.5 | 10.8 | ↑1.4 |
| | FARE[4] | 10.7 | 46.3 | 1.5 | 19.7 | 11.8 | 11.9 | 10.2 | 0.6 | 6.4 | 11.4 | 8.7 | 45.2 | 16.2 | 46.1 | 18.2 | |
| | LORE[4] | 17.8 | 54.2 | 2.8 | 27.4 | 16.8 | 14.4 | 10.0 | 0.6 | 8.0 | 16.4 | 11.7 | 48.4 | 25.5 | 56.1 | 22.5 | ↑4.3 |

## 6.4 Analysis: Additional Discussions and Ablation Studies

For further analysis, we present additional discussions and ablations through the following Q&As.

**Q: Why use a sample-specific tolerance margin $m(x)$ instead of a fixed one?**

**A:** The core of the LORE framework relies on a sample-specific tolerance margin, $m(x) = \rho\|\phi_{\theta_0}(x)\|^2$, which dynamically adjusts the allowable perturbation based on the feature magnitude of each input sample.

Table 4 compares LORE with the adaptive margin (LORE$_{m(x)}$, $\rho = 0.1$) against LORE with several fixed margins ($\rho = 0.6$, 0.8, and 1.5) as well as the FARE baseline. Experiments were conducted using ViT-B/32 CLIP models fine-tuned for 5 epochs on ImageNet-100.

The results show that fixed margins are either *overly restrictive*, with $\rho = 0.6$ severely degrading $\varepsilon = 4$ robustness, or *overly loose*, with $\rho = 1.5$ leading to performance similar to FARE. In contrast, the adaptive margin LORE$_{m(x)}$ consistently achieves the best robust accuracy, validating our design choice.

Further discussion and results are in Appendix J.6.

Table 4: Ablation on Adaptive Margin $m(x)$. Adversarial Accuracy is evaluated at the same $\varepsilon$ as training.

| Method | $\rho$ | Clean | Robust |
|---|---|---|---|
| LORE$^2{}_{m(x)}$ | 0.1 | 68.81 | **33.71** |
| LORE$^2$ | 0.6 | 70.12 | 18.65 |
| LORE$^2$ | 0.8 | 69.65 | 29.45 |
| LORE$^2$ | 1.5 | 63.45 | 25.47 |
| FARE$^2$ | — | 62.23 | 23.15 |
| LORE$^4{}_{m(x)}$ | 0.1 | 66.91 | **17.58** |
| LORE$^4$ | 0.6 | 67.21 | 1.23 |
| LORE$^4$ | 0.8 | 64.32 | 14.68 |
| LORE$^4$ | 1.5 | 55.24 | 11.74 |
| FARE$^4$ | — | 53.68 | 12.87 |

**Q: Can LORE's proximity principle be extended to supervised adversarial fine-tuning?**

**A:** We found that LORE's core principle of using proximity constraints to maintain clean performance during adversarial fine-tuning, is transferable and highly effective in the supervised setting. This demonstrates the method's broader applicability beyond its original unsupervised design.

We applied LORE's embedding-space proximity regularization ($\ell_2$) to TeCoA [Mao et al., 2023], creating the combined method TeCoA + $\ell_2$. We also explored a variant constraining the KL divergence of the model's output distribution (TeCoA + KL).

Table 5 presents the results, showing that TeCoA + $\ell_2$ consistently achieves the highest clean accuracy, with massive gains at high adversarial training budgets. Experiments were performed using ViT-B/32 CLIP models fine-tuned for 5 epochs on ImageNet-100. This validates that proximity-based constraints, implemented via LORE's Lagrangian framework, are a generally effective mechanism for stabilizing and improving both clean and robust performance in diverse adversarial fine-tuning scenarios.

Further discussion and results are in Appendix J.7.

Table 5: Extension to Supervised Fine-Tuning. Adversarial Accuracy is evaluated at the same $\varepsilon$ as training.

| Method | Clean | Robust |
|---|---|---|
| TeCoA$^2$ | 60.04 | 35.94 |
| **TeCoA$^2$ + $\ell_2$** | **75.97** | 39.12 |
| TeCoA$^2$ + KL | 66.50 | **39.53** |
| TeCoA$^4$ | 49.55 | 19.8 |
| **TeCoA$^4$ + $\ell_2$** | **73.12** | **49.12** |
| TeCoA$^4$ + KL | 56.01 | 22.18 |
| TeCoA$^8$ | 30.22 | 2.35 |
| **TeCoA$^8$ + $\ell_2$** | **67.23** | **4.23** |
| TeCoA$^8$ + KL | 38.06 | 2.82 |

## 7 Conclusion, Limitations and Future work

*Summary.* We proposed LORE, an unsupervised adversarial fine-tuning framework that enhances the robustness of visual encoders while preserving nominal performance without relying on heuristic loss weighting. Extensive experiments show that LORE consistently outperforms FARE, particularly under stronger attacks, achieving superior robustness with better clean data accuracy. LORE's robust visual encoders improve reliability in critical applications, fostering trust in AI. Maintaining high, clean data accuracy ensures effective performance in standard operational environments.

*Limitations and future work.* While LORE is effective, it has several limitations that suggest directions for future work. (1) A deeper theoretical analysis of the trade-offs in unsupervised adversarial fine-tuning is needed; our constrained framework provides a useful starting point. (2) The effectiveness and performance ceiling of LORE, which relies on the pretrained model as a fixed anchor ($\phi_{\theta_0}$), are inherently limited by the quality and fidelity of that frozen reference model. (3) Our use of a neural network to model Lagrange multipliers is heuristic; better parameterizations could improve efficiency and reduce duality gaps. (4) While we adopt Lagrangian duality with manageable gaps, alternative constrained optimization techniques may offer stronger guarantees. Future work could also explore supervised LORE variants.

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

# Part I

# Appendix

## Table of Contents

# A    Proof of Theorem 6.2

We define the following quantities:

$$R = \min_{\phi \in \mathcal{H}} \ell_{\text{adv}}(\phi; \phi_0), \quad R_\rho = \min_{\phi \in \mathcal{H}_\rho} \ell_{\text{adv}}(\phi; \phi_0).$$

And

$$\phi_\rho^* = \operatorname*{argmin}_{\phi \in \mathcal{H}_\rho} \mathbb{E}_{x \sim \mathcal{D}} \left[ \ell_{\text{adv}}(\phi, x; \phi_{\theta_0}) \right] = \operatorname*{argmin}_{\phi \in \mathcal{H}_\rho} \ell_{\text{adv}}(\phi; \phi_{\theta_0}),$$

$$\text{where} \quad \ell_{\text{adv}}(\phi, x; \phi_{\theta_0}) \triangleq \max_{\delta \in \Delta} d(\phi(x + \delta), \phi_{\theta_0}(x)). \tag{9}$$

Since $\mathcal{H}_\rho \subset \mathcal{H}$, we have $R_\rho \geq R$. The sub-optimality gap is bounded by:

$$0 \leq R_\rho - R \leq \sqrt{k} \cdot \text{Lipschitz}(\phi_\rho^* - \phi^*)\varepsilon + \|\phi_\rho^* - \phi^*\| \leq \sqrt{k}(L_\rho^* + L')\varepsilon + \|\phi_\rho^* - \phi^*\|.$$

*Proof.* Let $\delta_1^* = \arg\max_{\delta \in \Delta} d(\phi_1(x + \delta), \phi_0(x))$ denote the perturbation that maximizes the adversarial loss for model $\phi_1$ relative to the reference model $\phi_0$. For any two models $\phi_1, \phi_2$, we aim to bound the difference in their adversarial losses, considering assumption 6.1:

$$|\ell_{\text{adv}}(\phi_1, x; \phi_0) - \ell_{\text{adv}}(\phi_2, x; \phi_0)| \tag{10}$$

$$= \left| \max_{\delta \in \Delta} d(\phi_1(x + \delta), \phi_0(x)) - \max_{\delta \in \Delta} d(\phi_2(x + \delta), \phi_0(x)) \right| \tag{11}$$

$$\leq |d(\phi_1(x + \delta_1^*), \phi_0(x)) - d(\phi_2(x + \delta_1^*), \phi_0(x))| \tag{12}$$

$$\leq \|\phi_1(x + \delta_1^*) - \phi_2(x + \delta_1^*)\|. \tag{13}$$

Alternatively, we set the Lipschitz constant to 1 for brevity; however, it is worth noting that in general, this constant appears explicitly in the bound and should be carried through the analysis. Note that without loss of generality, we have assumed that $\max_{\delta \in \Delta} d(\phi_1(x + \delta), \phi_0(x)) - \max_{\delta \in \Delta} d(\phi_2(x + \delta), \phi_0(x)) \geq 0$. If that is not the case, then we have to replace $\delta_1^*$ with $\delta_2^* = \arg\max_{\delta \in \Delta} d(\phi_2(x + \delta), \phi_0(x))$ throughout the proof. We now decompose this difference using the triangle inequality:

$$\|\phi_1(x + \delta_1^*) - \phi_2(x + \delta_1^*)\| \tag{14}$$

$$= \|(\phi_1(x + \delta_1^*) - \phi_1(x)) - (\phi_2(x + \delta_1^*) - \phi_2(x)) + (\phi_1(x) - \phi_2(x))\| \tag{15}$$

$$\leq \|(\phi_1(x + \delta_1^*) - \phi_2(x + \delta_1^*)) - (\phi_1(x) - \phi_2(x))\| + \|\phi_1(x) - \phi_2(x)\|. \tag{16}$$

Assuming that the function $\phi_1 - \phi_2$ is Lipschitz continuous with constant $L$, we obtain:

$$\|(\phi_1(x + \delta_1^*) - \phi_2(x + \delta_1^*)) - (\phi_1(x) - \phi_2(x))\| \leq L\|\delta_1^*\|. \tag{17}$$

Assuming $\delta_1^* \in \mathbb{R}^k$ with $\|\delta_1^*\|_2 \leq \varepsilon\sqrt{k}$—which corresponds to assuming that $d$ is Lipschitz with respect to the Euclidean norm, and that $\varepsilon$ is bounded in the $\ell_\infty$ norm—we have:

$$|\ell_{\text{adv}}(\phi_1, x; \phi_0) - \ell_{\text{adv}}(\phi_2, x; \phi_0)| \leq L\sqrt{k}\varepsilon + \|\phi_1(x) - \phi_2(x)\|. \tag{18}$$

We now extend this pointwise bound to the expected adversarial loss:

$$|\ell_{\text{adv}}(\phi_1; \phi_0) - \ell_{\text{adv}}(\phi_2; \phi_0)| = |\mathbb{E}_x[\ell_{\text{adv}}(\phi_1, x; \phi_0) - \ell_{\text{adv}}(\phi_2, x; \phi_0)]| \tag{19}$$

$$\leq \mathbb{E}_x\left[|\ell_{\text{adv}}(\phi_1, x; \phi_0) - \ell_{\text{adv}}(\phi_2, x; \phi_0)|\right] \tag{20}$$

$$\leq \mathbb{E}_x\left[L\sqrt{k}\varepsilon + \|\phi_1(x) - \phi_2(x)\|\right] \tag{21}$$

$$= L\sqrt{k}\varepsilon + \|\phi_1 - \phi_2\|, \tag{22}$$

where $\|\phi_1 - \phi_2\|$ denotes the expected difference in their outputs over the input distribution.

Similarly, setting $\varepsilon = 0$, the adversarial loss reduces to the clean loss, yielding a corresponding bound:

$$|\ell_{\text{clean}}(\phi_1, \phi_0) - \ell_{\text{clean}}(\phi_2, \phi_0)| \leq \|\phi_1 - \phi_2\|.$$

Finally, applying this to the case $\phi_1 = \phi_\rho^*$ and $\phi_2 = \phi^*$, we obtain the desired bound on the deviation in adversarial loss due to constraining the hypothesis space. $\square$

# B  The Parametrization Gap

In this paper, we solved our constrained optimization problem, by optiming its dual problem, and using neural networks for the lagrangian multipliers. particularly solving:

$$d^* = \max_{\psi \in \Psi} \min_{\theta \in \Theta} \mathbb{E}[\max_{\delta \in \Delta} d(\phi_\theta(x + \delta), \phi_0(x)) + \lambda_\psi(x)(d(\phi_\theta(x), \phi_0(x)) - \rho m(x))]$$

Instead of the original constrained optimization problem. in this section we are interested in deriving some bounds on the duality gap, following the proof from Robey et al. [2021].

*Proof.*

**Assumption B.1.** For all $g \in \text{conv}(\mathcal{H})$, there exists $\tilde{\theta} \in \Theta$ such that

$$\|\phi_{\tilde{\theta}} - g^*\| \leq \eta,$$

where $\eta > 0$ is a sufficiently small constant.

First, ignoring the parametrization of $\lambda$, and assuming $\lambda$ can be any function from $\Lambda = \{\lambda : \mathcal{X} \to \mathbb{R}_+\}$, we consider the Lagrangian:

$$L(\phi, \lambda) = \mathbb{E}\left[\max_{\delta \in \Delta} d(\phi(x + \delta), \phi_0(x)) + \lambda(x)(d(\phi(x), \phi_0(x)) - \rho m(x))\right]$$

$$d^* = \sup_{\lambda \in \Lambda} \inf_{\theta \in \Theta} L(\phi_\theta, \lambda)$$

Now consider the original problem:

$$p^* = \inf_{\theta \in \Theta} \mathbb{E}_{x \sim \mathcal{D}}\left[\max_{\delta \in \Delta} d(\phi_\theta(x + \delta), \phi_0(x))\right],$$

$$\text{s.t.} \quad d(\phi_\theta(x), \phi_{\theta_0}(x)) \leq \rho m(x), \quad \text{for almost every } x \in \mathcal{X}$$

If the function class $\mathcal{H}$ parametrized by $\theta$ were convex, this would be a convex program. Since by definition, $\phi_0 \in \mathcal{H} = \{\phi_\theta : \theta \in \Theta\}$, there exists $\theta \in \Theta$ such that $d(\phi_\theta(x), \phi_0(x)) = 0 < \rho m(x)$ for all $x \in \mathcal{X}$. Thus, Slater's condition is satisfied.

Therefore, if $\mathcal{H}$ were convex, we would have strong duality, i.e., $p^* = d^*$. However, for most typical neural networks, $\mathcal{H}$ is non-convex. By weak duality, we always have: $d^* \leq p^*$. To derive a lower bound, consider the following problem for some positive constant $\eta > 0$:

$$\tilde{p}^* = \inf_{g \in \text{conv}(\mathcal{H})} \mathbb{E}_{x \sim \mathcal{D}}\left[\max_{\delta \in \Delta} d(g(x + \delta), \phi_0(x))\right],$$

$$\text{s.t.} \quad d(g(x), \phi_{\theta_0}(x)) \leq \rho m(x) - \eta, \quad \text{for almost every } x \in \mathcal{X}$$

This is now a convex program. Since $\phi_0$ itself satisfies $d(\phi_0(x), \phi_0(x)) = 0 < \rho m(x) - \eta$, Slater's condition is again satisfied. Hence, strong duality holds, and the Lagrangian becomes:

$$\tilde{L}(g, \lambda) = \mathbb{E}\left[\max_{\delta \in \Delta} d(g(x + \delta), \phi_0(x)) + \lambda(x)(d(g(x), \phi_0(x)) - \rho m(x) + \eta)\right] = L(g, \lambda) + \eta\mathbb{E}[\lambda(x)]$$

Thus,

$$\tilde{p}^* = \sup_{\lambda \in \Lambda} \inf_{g \in \text{conv}(\mathcal{H})} \tilde{L}(g, \lambda)$$

Assuming the infimum and supremum are attained at $g^*$ and $\tilde{\lambda}^*$, we have:

$$d^* - \sup_{\lambda \in \Lambda} \inf_{\theta \in \Theta} L(\phi_\theta, \lambda) \geq \inf_{\theta \in \Theta} L(\phi_\theta, \tilde{\lambda}^*) = \inf_{\phi \in \mathcal{H}} L(\phi, \tilde{\lambda}^*) \geq \inf_{g \in \text{conv}(\mathcal{H})} L(g, \tilde{\lambda}^*)$$

Using the relation between $\tilde{L}$ and $L$, we obtain:

$$d^* \geq \inf_{g \in \text{conv}(\mathcal{H})} L(g, \tilde{\lambda}^*) = \inf_g \left[\tilde{L}(g, \tilde{\lambda}^*) - \eta\mathbb{E}[\tilde{\lambda}(x)]\right] = \tilde{L}(g^*, \tilde{\lambda}^*) - \eta\mathbb{E}[\tilde{\lambda}(x)] = \tilde{p}^* - \eta\mathbb{E}[\tilde{\lambda}(x)]$$

Since $g^*$ is strictly feasible for the relaxed constraint, the complementary slackness condition implies $\tilde{\lambda}^*(x) = 0$ for almost every $x$. Therefore:

$$d^* \geq \tilde{p}^* = \mathbb{E}\left[\max_{\delta \in \Delta} d(g^*(x + \delta), \phi_0(x))\right] = \ell_{\text{adv}}(g^*, \phi_0)$$

Using the suboptimality bound from Lemma A, we have:

$$|\ell_{\text{adv}}(g^*, \phi_0) - \ell_{\text{adv}}(\phi_{\tilde{\theta}}, \phi_0)| \leq L\sqrt{k}\varepsilon + \|g^* - \phi_{\tilde{\theta}}\|, \quad |\ell_{\text{clean}}(g^*, \phi_0) - \ell_{\text{clean}}(\phi_{\tilde{\theta}}, \phi_0)| \leq \|g^* - \phi_{\tilde{\theta}}\|$$

Since $g^*$ is feasible in the relaxed problem with the stricter constraint $d(g^*(x), \phi_0(x)) \leq \rho m(x) - \eta$, for some $\tilde{\theta}^* \in \Theta$ approximating $g^*$ such that $\|g^* - \phi_{\tilde{\theta}^*}\| < \eta$, the function $\phi_{\tilde{\theta}^*}$ is feasible in the original problem, because for almost every $x \in \mathcal{X}$

$$\ell_{\text{clean}}(\phi_{\tilde{\theta}^*}(x), \phi_0(x)) \leq \ell_{\text{clean}}(g^*(x), \phi_0(x)) + \|g^* - \phi_{\tilde{\theta}^*}^*\| \leq \rho m(x) - \eta + \|g^* - \phi_{\tilde{\theta}^*}\| < \rho m(x).$$

Since $p^*$ is the minimum over all feasible $\theta \in \Theta$, it follows that $p^* \leq \ell_{\text{adv}}(\phi_{\tilde{\theta}^*}, \phi_0)$.

$$d^* \geq \ell_{\text{adv}}(g^*, \phi_0) \geq \ell_{\text{adv}}(\phi_{\tilde{\theta}^*}, \phi_0) - L\sqrt{k}\varepsilon - \|g^* - \phi_{\tilde{\theta}^*}\| \geq \ell_{\text{adv}}(\phi_{\tilde{\theta}^*}, \phi_0) - L\sqrt{k}\varepsilon - \eta$$

Therefore, since $\theta^*$ is the optimal solution to the original problem:

$$d^* \geq p^* - L\sqrt{k}\varepsilon - \eta$$

Finally, we note that although we have treated the dual space as the full infinite-dimensional set $\Lambda = \{\lambda : \mathcal{X} \to [0, \infty)\}$, in this work we have restricted $\lambda$ to a finite-dimensional family $\{\lambda_\omega\}_{\omega \in \Omega} \subset \Lambda$ that uniformly approximates its elements. Concretely, if for every $\lambda \in \Lambda$ there exists $\omega \in \Omega$ with $\|\lambda - \lambda_\omega\|_{L^1(\mathcal{D})} \leq \xi$, then replacing

$$\sup_{\lambda \in \Lambda} \inf_{\phi \in \mathcal{H}} L(\phi, \lambda) \quad \text{by} \quad \sup_{\omega \in \Omega} \inf_{\phi \in \mathcal{H}} L(\phi, \lambda_\omega)$$

only incurs an arbitrarily small error $O(\xi)$. All weak-duality arguments carry over immediately, and under Slater's condition the resulting strong-duality statement remains valid up to this negligible approximation.

One viewpoint is that limiting the expressivity of $\lambda_\omega$ through parametrization effectively relaxes the constraints, as the network cannot fully ensure the constraints are met. As an extreme case, consider when $\lambda_\omega(x) = \omega$ for all $x \in \mathcal{X}$. In this case:

$$\lambda_\omega(x) \equiv \omega, \quad \omega \geq 0.$$

Then the (relaxed) Lagrangian becomes

$$L(\phi, \omega) = \mathbb{E}\left[\max_{\delta \in \Delta} d(\phi(x + \delta), \phi_0(x))\right] + \omega\, \mathbb{E}\left[d(\phi(x), \phi_0(x)) - \rho\, m(x)\right].$$

Optimizing first over $\phi$ (so that $\mathbb{E}[\max_\delta d]$ is fixed) and then taking the supremum over $\omega \geq 0$ forces

$$\mathbb{E}\left[d(\phi(x), \phi_0(x))\right] - \rho\, \mathbb{E}[m(x)] \leq 0,$$

otherwise $L(\phi, \omega) \to +\infty$ as $\omega \to +\infty$. In other words, the constant-$\lambda$ relaxation exactly enforces

$$\mathbb{E}\left[d(\phi(x), \phi_0(x))\right] \leq \rho\, \mathbb{E}[m(x)].$$

Thus, by limiting the expressivity of $\lambda$, we move from the original per-sample constraint

$$d(\phi(x), \phi_0(x)) \leq \rho\, m(x) \quad \forall x,$$

to the weaker but still meaningful average constraint

$$\mathbb{E}\left[d(\phi(x), \phi_0(x))\right] \leq \rho\, \mathbb{E}[m(x)].$$

$\square$

# C   Additional Experimental Details

In this appendix, we provide further experimental details beyond those given in the main text. Experiments were conducted using 8 NVIDIA HGX H100 80GB GPUs.

**Training hyperparameters.** We report below the training settings used across all experiments. Unless otherwise noted, all models were trained using AdamW with a weight decay of $1 \times 10^{-4}$, a cosine learning rate scheduler, and adversarial training with PGD (10 iterations, step size $\varepsilon/4$) under an $\ell_\infty$ constraint. Each $\lambda$ network used the 2-layer `linear_mlp` architecture, with a hidden dimension of 512, and was optimized via $K = 5$ inner primal updates with learning rate $5 \times 10^{-4}$. More experimental details are provided in Table 6. Additional information about all figures and tables in the paper is summarized in Table 7.

Table 6: Training hyperparameters for all models trained with LORE. All models trained with FARE use the same number of epochs and learning rate as the corresponding LORE setting.

| Model | Training Dataset | Epochs | Batch size (per device) | LR | $\rho$ | $K$-iter | $\lambda$ LR |
|---|---|---|---|---|---|---|---|
| | | | *LORE*[1] | | | | |
| CLIP ViT-B/32 | ImageNet | 2 | 448 | 1e-5 | 0.1 | 5 | 5e-4 |
| CLIP ViT-B/32 | ImageNet-100 | 5 | 448 | 1e-5 | 0.1 | 5 | 5e-4 |
| CLIP ViT-B/32 | CIFAR10 | 5 | 448 | 1e-5 | 0.01 | 5 | 5e-4 |
| | | | *LORE*[2] | | | | |
| CLIP ViT-B/16 | ImageNet-100 | 5 | 128 | 1e-5 | 0.1 | 5 | 5e-4 |
| CLIP ViT-B/32-LAION | ImageNet-100 | 5 | 448 | 1e-5 | 0.15 | 5 | 5e-4 |
| CLIP ConvNeXt-B | ImageNet-100 | 5 | 64 | 1e-5 | 0.15 | 5 | 5e-4 |
| CLIP ViT-B/32 | ImageNet | 2 | 448 | 1e-5 | 0.1 | 5 | 5e-4 |
| CLIP ViT-B/32 | ImageNet-100 | 5 | 448 | 1e-5 | 0.1 | 5 | 5e-4 |
| CLIP ViT-B/32 | CIFAR10 | 5 | 448 | 1e-5 | 0.01 | 5 | 5e-4 |
| | | | *LORE*[4] | | | | |
| CLIP ViT-B/16 | ImageNet-100 | 5 | 128 | 1e-5 | 0.2 | 5 | 5e-4 |
| CLIP ViT-B/32-LAION | ImageNet-100 | 5 | 448 | 1e-5 | 0.15 | 5 | 5e-4 |
| CLIP ConvNeXt-B | ImageNet-100 | 5 | 64 | 1e-5 | 0.15 | 5 | 5e-4 |
| DINOv2 ViT-S/14 | ImageNet | 2 | 128 | 1e-5 | 0.05 | 5 | 5e-4 |
| DINOv2 ViT-B/14 | ImageNet | 1 | 64 | 1e-5 | 0.1 | 5 | 5e-4 |
| CLIP ViT-B/32 | ImageNet | 3 | 448 | 1e-5 | 0.1 | 5 | 5e-4 |
| CLIP ViT-B/32 | ImageNet-100 | 5 | 448 | 1e-5 | 0.1 | 5 | 5e-4 |
| CLIP ViT-B/32 | CIFAR10 | 5 | 448 | 1e-5 | 0.01 | 5 | 5e-4 |
| | | | *LORE*[6] | | | | |
| CLIP ViT-B/32 | ImageNet | 3 | 448 | 1e-5 | 0.2 | 5 | 5e-4 |
| CLIP ViT-B/32 | ImageNet-100 | 5 | 448 | 1e-5 | 0.1 | 5 | 5e-4 |
| | | | *LORE*[8] | | | | |
| CLIP ViT-B/16 | ImageNet-100 | 5 | 128 | 1e-5 | 0.2 | 5 | 5e-4 |
| CLIP ViT-B/32-LAION | ImageNet-100 | 5 | 448 | 1e-5 | 0.15 | 5 | 5e-4 |
| CLIP ConvNeXt-B | ImageNet-100 | 5 | 64 | 1e-5 | 0.15 | 5 | 5e-4 |
| DINOv2 ViT-S/14 | ImageNet | 2 | 128 | 1e-5 | 0.05 | 5 | 5e-4 |
| DINOv2 ViT-B/14 | ImageNet | 1 | 64 | 1e-5 | 0.1 | 5 | 5e-4 |
| CLIP ViT-B/32 | ImageNet | 3 | 448 | 1e-5 | 0.2 | 5 | 5e-4 |
| CLIP ViT-B/32 | ImageNet-100 | 5 | 448 | 1e-5 | 0.1 | 5 | 5e-4 |
| | | | *LORE*[10] | | | | |
| CLIP ViT-B/32 | ImageNet | 3 | 448 | 1e-5 | 0.2 | 5 | 5e-4 |
| CLIP ViT-B/32 | ImageNet-100 | 5 | 448 | 1e-5 | 0.1 | 5 | 5e-4 |
| | | | *LORE*[16] | | | | |
| DINOv2 ViT-S/14 | ImageNet | 2 | 128 | 1e-5 | 0.05 | 5 | 5e-4 |
| DINOv2 ViT-B/14 | ImageNet | 1 | 64 | 1e-5 | 0.1 | 5 | 5e-4 |

Table 7: Details of each Figure and Table used in the paper.

| Figure/Table | Model | Training Dataset | Additional Notes |
|---|---|---|---|
| Figure 1-a | FARE[2,4,6,8,10], LORE[10] | ImageNet-100 | Initial performance drop |
| Figure 1-b | FARE[2], LORE[2] | ImageNet-100 | Robustness–accuracy Pareto front |
| Figure 3 | LORE[2] | ImageNet-100 | Controllability of LORE |
| Figure 4-a | FARE[2], LORE[2] | ImageNet | OOD robustness |
| Figure 4-b | FARE[1,2,4,6,8,10], LORE[1,2,4,6,8,10] | ImageNet | Effect on image embedding interpretability |
| Figure 5 | FARE[1,2,4,6,8,10], LORE[1,2,4,6,8,10] | ImageNet | Accuracy & robust accuracy across $\varepsilon$ |
| Figure 6 | LORE[4] | ImageNet | Fidelity analysis of LORE |
| Table 1 | FARE[2,4], LORE[2,4] | ImageNet-100 | In-domain image classification |
| Table 2 | FARE[4,8], LORE[4,8] | ImageNet-100 | In-domain image classification (DINOv2) |
| Table 3 | FARE[1,2,4], LORE[1,2,4] | ImageNet | Zero-shot image classification |

## D   Constraint Satisfaction in LORE

To further understand the behavior of constraint enforcement under varying adversarial budgets, we visualize the distribution of distances between clean embeddings and their corresponding pre-trained references throughout training for $\varepsilon = 1, 2,$ and $4$ in Figure 7. As we can observe, larger perturbation strengths lead to greater deviation from the pre-trained reference in the early stages of training. This early-stage divergence results in a more pronounced initial drop in the model's nominal performance. However, LORE is able to effectively regulate this deviation through its constraint-aware mechanism, gradually aligning the clean embeddings back within the $\rho$ threshold. This demonstrates the robustness and practicality of LORE in preserving clean performance even under severe adversarial training regimes.

In contrast, due to the lack of such constraint regulation in FARE, the distance between clean embeddings and pre-trained references cannot be reliably controlled. As a result, FARE experiences a catastrophic initial drop in nominal accuracy, particularly under larger perturbation budgets. This failure to maintain embedding fidelity further underscores the importance of the dual network in LORE for stabilizing the training process and preserving clean accuracy.

To better illustrate this behavior, all subfigures in Figure 7 show the distance distributions beginning from the 20th training iteration onward. These comparisons clearly highlight the contrast between LORE's effective enforcement of the proximity constraint and FARE's limited capability to manage deviation across increasing adversarial strengths.

## E   Generalization Gap in Adversarial Training

**Theorem E.1** (Generalization Gap in Adversarial Training). *It is well known that the generalization gap for a given loss function is upper bounded by complexity measures, giving rise to theoretical justifications of the bias-variance trade-off. Assuming bounded norm embeddings, i.e., $\|\phi(x)\|_2 \leq K$ for all $x, \phi$, we can see that the uniform loss bound $B$ satisfies:*

$$B := \sup_{\phi,x} |\ell(\phi, x)| = \sup_{\phi,x} \max_{\delta} \|\phi(x + \delta) - \phi_0(x)\|_2 \leq \sup_{\phi,x} \max_{\delta} [\|\phi(x + \delta)\|_2 + \|\phi_0(x)\|_2] \leq 2K.$$

*Therefore, with probability at least $1 - 2\delta$,*

$$\left| \mathbb{E}_x[\ell(\phi, x)] - \frac{1}{|\mathcal{D}|} \sum_{i=1}^{|\mathcal{D}|} \ell(\phi, x_i) \right| \leq 2\mathfrak{R}_n(\mathcal{L}_\mathcal{H}) + B\sqrt{\frac{\log(1/\delta)}{2|\mathcal{D}|}} \leq 2\mathfrak{R}_n(\mathcal{L}_\mathcal{H}) + K\sqrt{\frac{2\log(1/\delta)}{|\mathcal{D}|}},$$

*where $\mathcal{L}_\mathcal{H} = \{\ell(\phi, \cdot) \mid \phi \in \mathcal{H}\}$ is the loss class induced by hypothesis class $\mathcal{H}$, $\mathfrak{R}_n(\mathcal{L}_\mathcal{H})$ is the empirical Rademacher complexity of $\mathcal{L}_\mathcal{H}$, and $\sup_{\phi,x} |\ell(\phi, x)| \leq B$.*

In adversarial training, the loss class $\mathcal{L}_\mathcal{H}$ becomes extremely complex due to the inner $\max_\delta$ operation, leading to large Rademacher complexity $\mathfrak{R}_n(\mathcal{L}_\mathcal{H})$. This explains why adversarial training requires significantly more samples for generalization compared to standard training.

When we restrict to simpler hypothesis classes $\mathcal{H}_\rho \subset \mathcal{H}_{\text{org}}$ (through techniques like Lipschitz constraints or norm bounds), the Rademacher complexity decreases, potentially improving generalization.

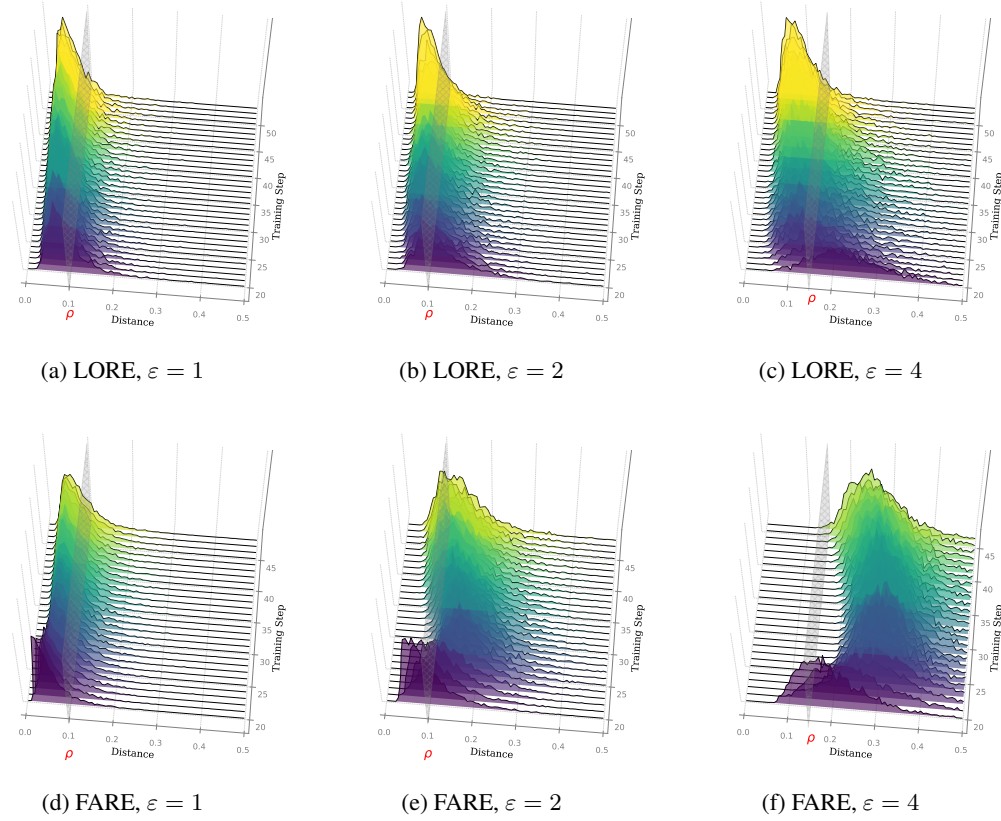

(a) LORE, $\varepsilon = 1$      (b) LORE, $\varepsilon = 2$      (c) LORE, $\varepsilon = 4$

(d) FARE, $\varepsilon = 1$      (e) FARE, $\varepsilon = 2$      (f) FARE, $\varepsilon = 4$

Figure 7: Constraint satisfaction comparison between LORE and FARE across adversarial training budgets $\varepsilon = 1, 2, 4$. **(Top):** LORE maintains strong proximity between clean embeddings and pre-trained references throughout training, with distances concentrating below the $\rho$ threshold. **(Bottom):** FARE exhibits weaker fidelity preservation and fails to effectively regulate distance under increasing adversarial strength.

However, the bounds we derived are notoriously crude; they fail to capture important phenomena like double descent and often dramatically overestimate the actual generalization gap in practice.

## F    Deviation Between Cosine Similarities

**Assumption F.1.** For each input $x$, let

$$u = \phi_{\mathrm{org}}(x), \quad \hat{u} = \phi_\theta(x),$$

and suppose

$$\|\hat{u} - u\|_2^2 \leq \rho \|u\|_2^2 \quad \text{for some } \rho \in [0, 1).$$

**Proposition F.2.** *Under the above assumption, for any nonzero $v \in \mathbb{R}^n$,*

$$\left| S_C(u, v) - S_C(\hat{u}, v) \right| \leq 2\sqrt{\rho}.$$

We show that enforcing

$$\|\phi_\theta(x) - \phi_{\mathrm{org}}(x)\|_2^2 \leq \rho \|\phi_{\mathrm{org}}(x)\|_2^2$$

implies a uniform bound on the change in cosine similarity to any fixed vector $v \in \mathbb{R}^n$.

*Proof.* Write

$$S_C(u, v) = \frac{v^T u}{\|v\| \|u\|}, \quad S_C(\hat{u}, v) = \frac{v^T \hat{u}}{\|v\| \|\hat{u}\|},$$

so

$$\left| S_C(u,v) - S_C(\hat{u},v) \right| = \left| \frac{v^T}{\|v\|} \left( \frac{u}{\|u\|} - \frac{\hat{u}}{\|\hat{u}\|} \right) \right| \leq \left\| \frac{u}{\|u\|} - \frac{\hat{u}}{\|\hat{u}\|} \right\|.$$

Now decompose

$$\frac{u}{\|u\|} - \frac{\hat{u}}{\|\hat{u}\|} = \left( \frac{u}{\|u\|} - \frac{\hat{u}}{\|u\|} \right) + \left( \frac{\hat{u}}{\|u\|} - \frac{\hat{u}}{\|\hat{u}\|} \right),$$

so by the triangle inequality,

$$\left\| \frac{u}{\|u\|} - \frac{\hat{u}}{\|\hat{u}\|} \right\| \leq \frac{\|u - \hat{u}\|}{\|u\|} + \|\hat{u}\| \left| \frac{1}{\|u\|} - \frac{1}{\|\hat{u}\|} \right|.$$

Since $\left| \|\hat{u}\| - \|u\| \right| \leq \|u - \hat{u}\|$ and $\|\hat{u}\| \leq \|u\| + \|u - \hat{u}\|$, one shows easily

$$\|\hat{u}\| \left| \frac{1}{\|u\|} - \frac{1}{\|\hat{u}\|} \right| \leq \frac{\|u - \hat{u}\|}{\|u\|}.$$

Hence

$$\left| S_C(u,v) - S_C(\hat{u},v) \right| \leq 2 \frac{\|u - \hat{u}\|}{\|u\|} \leq 2\sqrt{\rho},$$

as claimed. $\qquad\square$

**Remark.** In vision–language models one may take $v = \psi(t)$, the text embedding of prompt $t$, so the same bound guarantees $\left| S_C(\phi_{\mathrm{org}}(x), \psi(t)) - S_C(\phi_\theta(x), \psi(t)) \right| \leq 2\sqrt{\rho}$.)

## G   Impact of Dual Network on Model Performance

### G.1   Comparison of Alternative Architectures for the Dual Function

In Figure 8, we compare the clean and robust accuracy achieved by two different architectures used for the dual function: a simple scalar value and a network-based (sample-dependent) function, as adopted in the current LORE setting. While both configurations perform comparably in terms of clean accuracy, the network-based dual function generalizes substantially better on adversarial examples, leading to consistently higher robust accuracy throughout training. This highlights the importance of a flexible, sample-adaptive mechanism in effectively enforcing robustness constraints during adversarial fine-tuning.

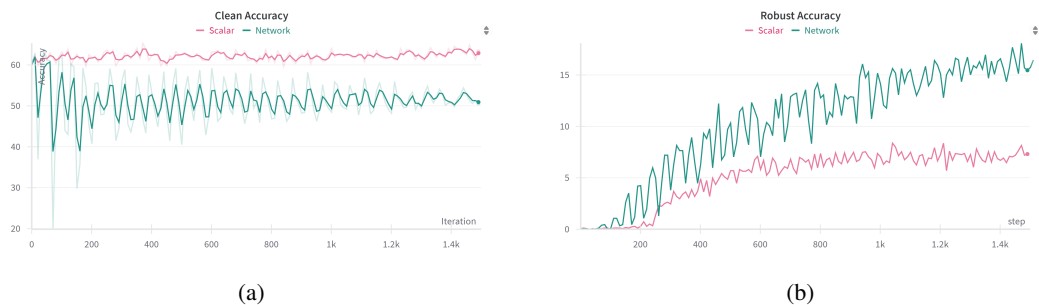

(a)                                           (b)

Figure 8: Comparison of clean and robust accuracy when using different architectures for the dual function in LORE. **(a):** Clean accuracy over training steps. **(b):** Robust accuracy over training steps. The *Network*-based dual function (sample-based) used in the current LORE setting leads to significantly higher robust accuracy compared to the *Scalar* baseline, while maintaining competitive clean accuracy.

**G.2    Effect of the Dual Network $\lambda_\omega(x)$ on Clean Accuracy**

Figure 9 illustrates the impact of the dual network $\lambda_\omega(x)$ on model performance. As shown, at the initial steps, higher values of $\lambda_\omega(x)$ help maintain the model's nominal performance, ensuring it performs well on clean data. In contrast, for FARE, due to the absence of such a proximity constraint during the early iterations, the model, lacking robustness, passes through suboptimal states, leading to a significant drop in nominal performance.

To further support this observation, we present comprehensive results in Figure 10 and Figure 11, showcasing the behavior of the dual network and its impact across different architectures. In Figure 10, experiments on DINOv2 models (base and small) demonstrate that LORE consistently achieves higher clean accuracy compared to FARE, especially in the early stages of training, while adaptively modulating $\lambda_\omega(x)$ to control constraint satisfaction. Similarly, Figure 11 reports the performance of ViT-B/16 and ConvNeXt-B models, confirming the effectiveness and generality of the dual network across various architectures and perturbation strengths. These results highlight that LORE's constraint-aware mechanism is stable, avoiding the sharp degradation commonly observed in FARE adversarial fine-tuning.

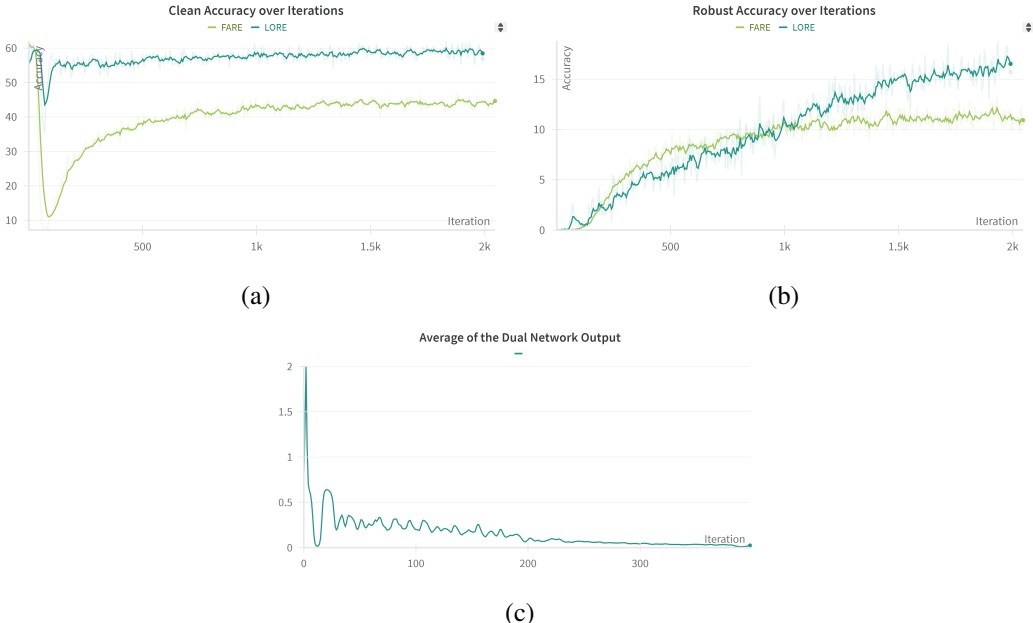

(a)                 (b)

(c)

Figure 9: Comparison of the performance of two methods and the output of the dual network. (a) Clean accuracies over iterations, (b) Robust accuracies over iterations, (c) Average output of the dual network $\lambda_\omega(x)$.

# H    Revisiting Embedding Models

**CLIP** [Radford et al., 2021]. A major part of our experiments builds upon CLIP, which consists of an image encoder $\phi$ and a text encoder $\psi$ that map images and text descriptions into a shared embedding space. For zero-shot classification, textual descriptions are typically formatted as `"This is a photo of a [CLS]''`, where `[CLS]` represents class labels. The probability of assigning an image $x$ to a class $\hat{y}$ is computed via a softmax over cosine similarities:

$$p(\hat{y} \mid x) = \frac{\exp(\cos(\psi(t_{\hat{y}}), \phi(x))/\tau)}{\sum_{j=1}^{K} \exp(\cos(\psi(t_j), \phi(x))/\tau)}. \tag{23}$$

where $\tau$ is a temperature parameter, and $K$ denotes the number of classes.

**DINOv2** [Oquab et al., 2024]. In addition to CLIP, we incorporate DINOv2, a powerful self-supervised visual transformer-based encoder, into our exploration of embedding models. While

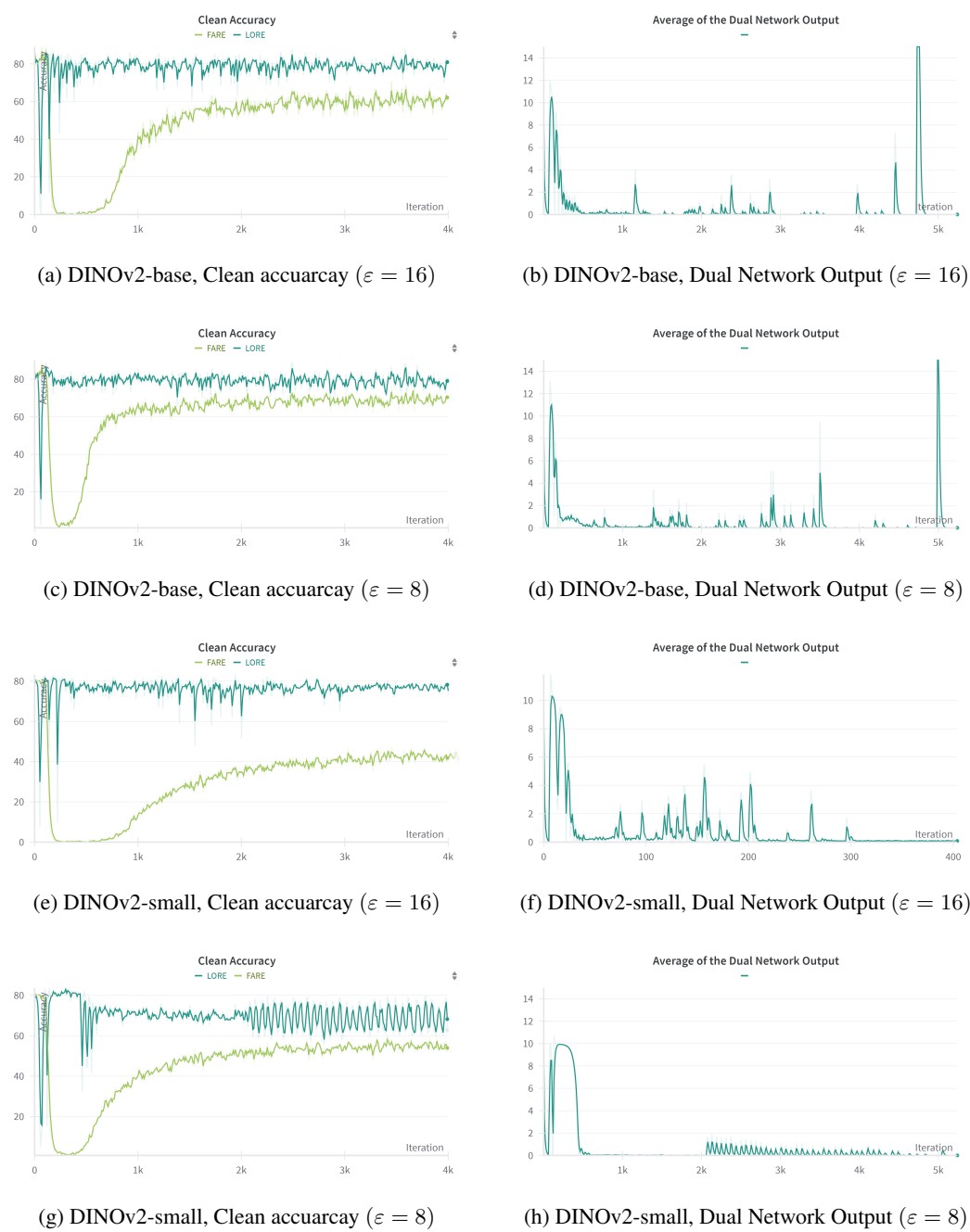

Figure 10: Comparison of LORE and FARE on DINOv2-base and DINOv2-small models under different adversarial budgets $\varepsilon \in \{8, 16\}$. **Left:** Clean accuracy over training iterations, illustrating LORE's superior stability and performance. **Right:** Average output of the dual network $\lambda_\omega(x)$ across iterations, highlighting how LORE dynamically adjusts its constraint enforcement.

CLIP provides a joint image-text embedding space, DINOv2 focuses solely on visual representation learning. This complementary perspective allows us to compare and leverage both multimodal and unimodal embedding paradigms.

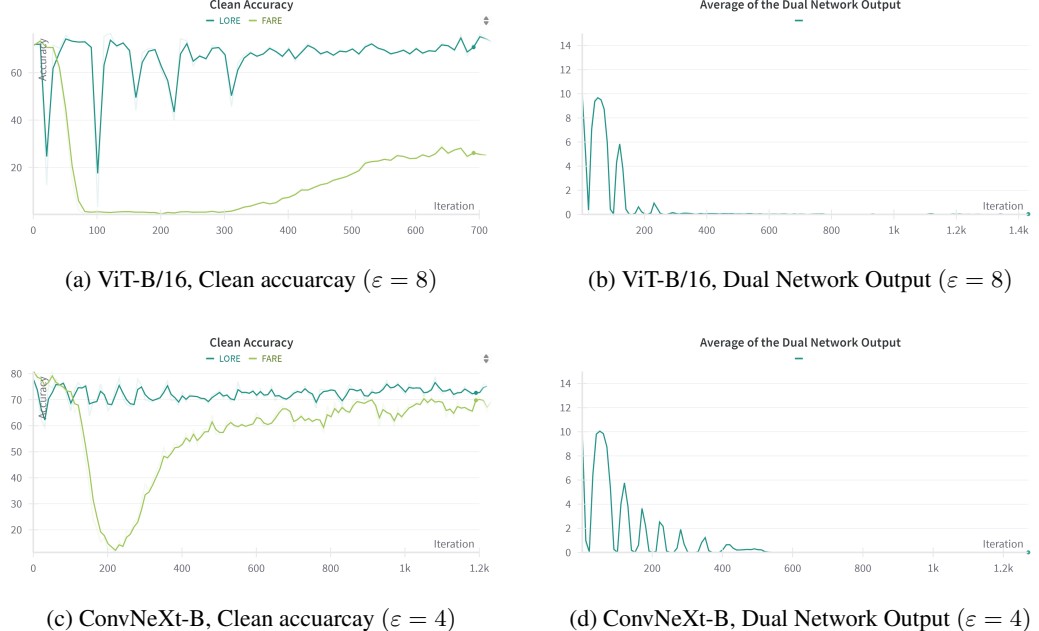

(a) ViT-B/16, Clean accuarcay ($\varepsilon = 8$)  (b) ViT-B/16, Dual Network Output ($\varepsilon = 8$)

(c) ConvNeXt-B, Clean accuarcay ($\varepsilon = 4$)  (d) ConvNeXt-B, Dual Network Output ($\varepsilon = 4$)

Figure 11: Comparison of LORE and FARE on ViT-B/16 and ConvNeXt-B models under adversarial fine-tuning. **Left:** Clean accuracy over training iterations ($\varepsilon = 8$ for ViT-B/16 and $\varepsilon = 4$ for ConvNeXt-B), showing LORE's improved stability and performance. **Right:** Average output of the dual network $\lambda_\omega(x)$, indicating LORE's dynamic constraint adjustment during training.

DINOv2 learns visual features by minimizing a cross-view prediction loss between student and teacher networks. Given $N$ image views, the loss is computed as:

$$\mathcal{L}_{\text{DINO}} = -\frac{1}{N} \sum_{i=1}^{N} \sum_{c=1}^{C} q_{ic} \log p_{ic}, \tag{24}$$

where $q_{ic}$ and $p_{ic}$ are the teacher and student probabilities for class $c$ and view $i$, respectively, and $C$ is the number of output dimensions.

Our broader work centers around embedding models, with a primary emphasis on CLIP, while also investigating the capabilities and representations of models like DINOv2. In general, we found that the DINOv2 model has much richer image embeddings than CLIP and can achieve much higher robustness over the same perturbation and dataset.

## I  Impact of $K$ on Model Performance

LORE alternates between $K$ steps of updating the primal encoder and one step of updating the dual network. In this section, we study the effect of the hyperparameter $K$ on final performance. As shown in Table 8, increasing $K$ leads to improved clean accuracy (Acc) and robust accuracy (RAcc), particularly when moving from $K = 1$ to $K = 3$ or $5$. This demonstrates that more frequent primal updates between dual updates help stabilize training and improve performance. Based on this observation, we choose $K = 5$ as the default in our final LORE implementation. For further discussion on how $K$ impacts computational cost and training time, see Appendix K.

## J  Additional Experimental Results

In this section, we present additional experiments to further validate the robustness and generalization capabilities of our proposed method. These evaluations span multiple settings, including black-box adversarial attacks (e.g., Square Attack), Gaussian noise corruption, in-domain and zero-shot classification, and out-of-distribution (OOD) robustness. By comparing against the FARE baseline

Table 8: Effect of the $K$ hyperparameter on model performance. Clean accuracy (Acc) and robust accuracy (RAcc) (%) are reported for various values of $K$. Results are based on ViT-B/32 trained with LORE[2] evaluated on ImageNet-100 under $\varepsilon = 2/255$ APGD attack.

| K | 1 | 2 | 3 | 5 | 7 | 10 |
|---|---|---|---|---|---|---|
| **Acc (%)** | 64.11 | 66.43 | 64.58 | 60.19 | 59.96 | 56.46 |
| **RAcc (%)** | 9.54 | 13.78 | 19.48 | 27.01 | 31.81 | 39.71 |

Table 9: Evaluation on Square Attack, a Black-Box attacks, averaged over the previous mentioned 13 zero-shot datasets

| Method | Backbone | Clean | $\varepsilon = 1$ | $\varepsilon = 2$ | $\varepsilon = 4$ | $\varepsilon = 6$ |
|---|---|---|---|---|---|---|
| FARE[4] | ViT-B/32 | 42.6 | 40.0 | 36.4 | 30.0 | 23.6 |
| LORE[4] | ViT-B/32 | **50.1** | **43.9** | **40.3** | **33.5** | **27.1** |

across diverse conditions and datasets, we demonstrate that LORE consistently achieves superior performance, particularly under challenging threat models and distributional shifts.

## J.1 Square Attack Evaluation

In this section, we evaluate the robustness of our fine-tuning approach against a black-box adversarial attack known as the Square Attack [Andriushchenko et al., 2020]. Unlike gradient-based methods, Square Attack operates without access to model gradients and perturbs the input using a query-efficient, score-based strategy. This makes it a strong candidate for evaluating real-world robustness where white-box access is not feasible. We conduct experiments on LORE[4], which is adversarially fine-tuned on ImageNet, to assess how well the model generalizes to such black-box settings. The results, summarized in Table 9, show that our method consistently outperforms the baseline under this challenging threat model.

## J.2 Evaluation Under Gaussian Noise Corruption

To further assess the robustness of our method, we evaluate the performance of LORE[4] and FARE[4] under varying levels of Gaussian noise corruption. We use the ViT-B/32 CLIP model, with both methods fine-tuned on ImageNet. As shown in Figure 13, LORE[4] consistently maintains higher accuracy than FARE[4] across a wide range of noise strengths ($\sigma$), especially in low to moderate noise settings. The right subplot illustrates the accuracy gap, highlighting LORE[4]'s robustness advantage up to $\sigma = 40$, beyond which the performance of both models converges as the corruption becomes extreme. This evaluation further supports the generalization capabilities of our method in the presence of unseen corruptions. Figure 12 provides a visual illustration of how a single image degrades under increasing levels of Gaussian noise.

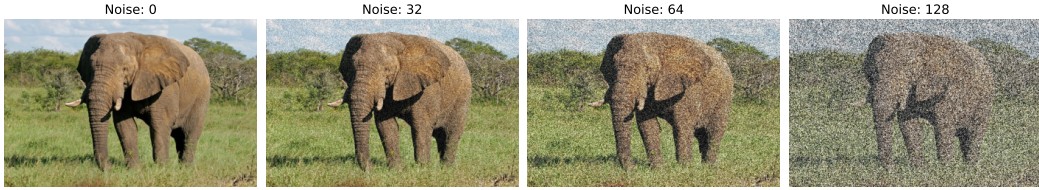

Figure 12: Visualization of a single image under increasing levels of Gaussian noise ($\sigma = 0, 32, 64, 128$). This figure helps set reasonable expectations for model performance by illustrating how perceptual degradation progresses with noise intensity.

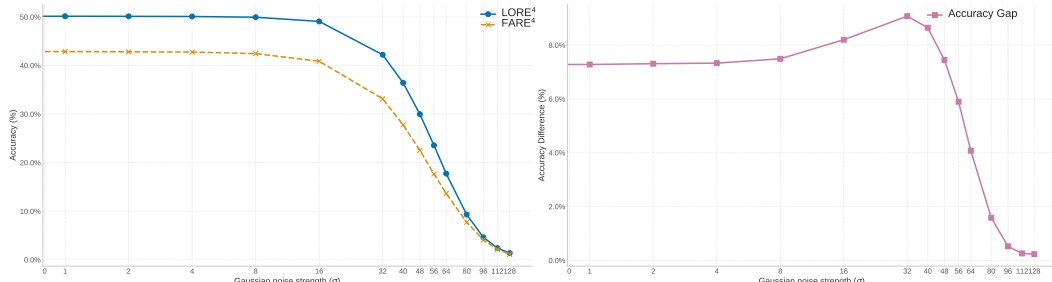

Figure 13: Robustness evaluation under Gaussian noise corruption. **Left:** Classification accuracy of LORE[4] and FARE[4] under increasing Gaussian noise strength ($\sigma$). LORE[4] consistently outperforms FARE[4], particularly in moderate noise regimes. **Right:** Accuracy gap between LORE[4] and FARE[4], showing a stable and significant margin up to $\sigma = 40$, after which the gap decreases as both models degrade under extreme noise conditions.

Table 10: Clean and adversarial accuracy for in-domain image classification on ImageNet-100 across different CLIP vision encoders, evaluated using the APGD attack.

| Method | Backbone | Clean | $\varepsilon = 1$ | $\varepsilon = 2$ | $\varepsilon = 4$ | $\varepsilon = 8$ |
|---|---|---|---|---|---|---|
| FARE[8] | ViT-B/16 | 26.5 | 20.4 | 17.0 | 10.3 | 2.3 |
| LORE[8] | ViT-B/16 | **70.5** | **53.6** | **48.5** | **37.8** | **17.8** |
| FARE[8] | ViT-B/32 LAION | 17.0 | 11.3 | 7.3 | 3.16 | 0.40 |
| LORE[8] | ViT-B/32 LAION | **28.2** | **12.1** | **10.0** | **6.54** | **3.51** |
| FARE[8] | ConvNeXt-B | 61.6 | 55.3 | 48.5 | 35.7 | 43.4 |
| LORE[8] | ConvNeXt-B | **72.2** | **56.2** | **49.1** | **38.3** | **47.2** |

### J.3  In-domain Image Classification

Table 10 presents a comparison of clean and adversarial accuracy across various CLIP-based vision backbones, all trained with $\varepsilon = 8$, on the ImageNet-100 dataset under the APGD attack.

### J.4  Zero-shot Image Classification

Table 11 presents the results of different settings for zero-shot image classification using the ViT-B/32 CLIP model, highlighting the superiority of our method over the FARE baseline. Additionally, for a more challenging scenario, Table 12 reports model performance under high-intensity adversarial attacks, further demonstrating the resilience of our approach. These tables serve as the complete version of the results summarized in the main paper.

### J.5  Out-of-Distribution Robustness

As shown in Figure 14, increasing the training perturbation strength leads to greater degradation in out-of-distribution (OOD) robustness across common corruptions in ImageNet-C. Despite this trend, models trained with LORE consistently exhibit better robustness compared to those trained with the FARE method, highlighting LORE's superior generalization under distributional shifts.

### J.6  Ablation on Sample-Specific Margin $m(x)$

The choice of a sample-specific tolerance margin, $m(x) = \rho \|\phi_{\theta_0}(x)\|^2$, is a crucial design decision in the LORE framework.

Using a sample-invariant tolerance ($\rho_{\text{fixed}}$), where the constraint is fixed across all data points ($d(\phi_\theta(x), \phi_{\theta_0}(x)) \leq \rho_{\text{fixed}}$), offers an alternative but is challenging. It requires extensive, task-dependent and model-dependent hyperparameter tuning that severely limits its practical utility and generalization ability.

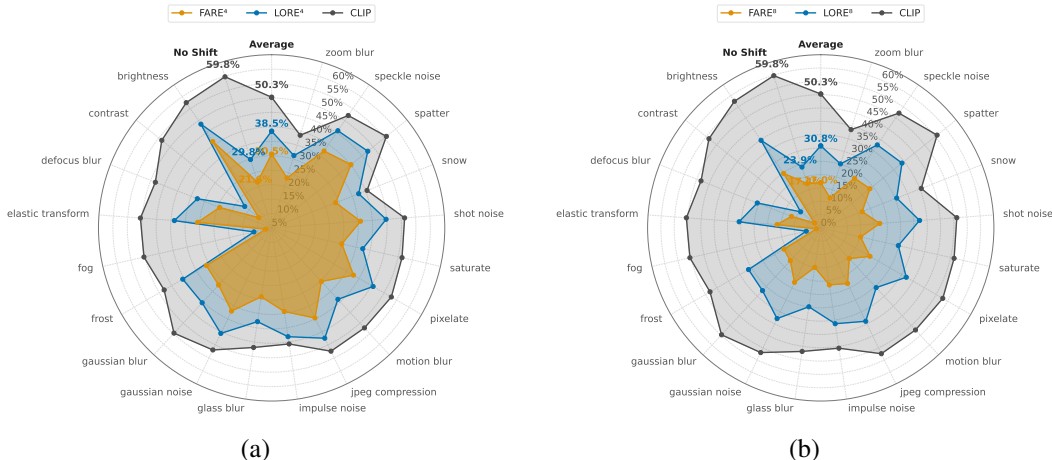

(a)                    (b)

Figure 14: Robustness to common corruptions on ImageNet-C as an OOD evaluation for models trained with (a) $\varepsilon = 4$, and (b) $\varepsilon = 8$. As we can observe, as the training perturbation strength increases, the degradation in OOD robustness also increases. Nevertheless, LORE consistently shows lower degradation compared to models trained with the FARE method.

By contrast, the sample-specific margin provides an adaptive, scale-invariant tolerance tied to the pre-trained embedding's norm. This design choice reduces manual tuning, preserves semantic structure by relating the constraint geometrically to the cosine similarity of embeddings, and ultimately offers greater stability.

**Experimental Comparison.** To validate this choice, we conducted experiments using ViT-B/32 CLIP models fine-tuned for 5 epochs on ImageNet-100. We compare the adaptive margin (LORE$_{m(x)}$) against several empirically chosen fixed tolerances ($\rho = 0.6, 0.8, 1.5$). The superscript ($\varepsilon$) denotes the training perturbation budget.

As shown in Table 13, fixed tolerances of $0.6$ and $1.5$ lead to suboptimal performance, demonstrating that a fixed margin is either overly restrictive (degrading robustness) or overly loose (causing behavior similar to FARE). The sample-specific margin (LORE$_{m(x)}$) consistently achieves the best overall Average Robust Accuracy, validating its superior stability and efficacy across different training settings.

### J.7 LORE in Supervised Adversarial Fine-Tuning

Given an image-label pair $(x, y)$, TeCoA [Mao et al., 2023] optimizes a contrastive loss to align adversarial image embeddings with text embeddings. The loss function is defined as:

$$L_{\text{sup}}(x, y; \theta) = \max_{\delta \in \Delta} - \sum_i \left[ y_i \log \frac{\exp(\cos(z, \psi(t_i))/\tau)}{\sum_k \exp(\cos(z, \psi(t_k))/\tau)} \right], \qquad (25)$$

where $z = \phi_\theta(x + \delta)$ is the adversarial image embedding and $\psi(t_i)$ is the text embedding. Here, $\psi(t_i)$ corresponds to the text embedding of the $i$-th class, where each class label is inserted into a fixed natural language template (e.g., "This is a photo of {class}").

We incorporated our proximity-based regularizer into TeCoA, forming a combined method (TeCoA + $l_2$), where $l_2$ denotes the $\ell_2$-based embedding proximity constraint:

$$\max_{\omega \in \Omega} \min_{\theta \in \Theta} E_{x, y \sim D} \left[ L_{\text{sup}}(x, y; \theta) + \lambda_\omega(x) \big( |\phi_\theta(x) - \phi_0(x)|_2^2 - \rho |\phi_0(x)|_2^2 \big) \right]$$

**Distribution-Level Constraint.** We also explored a probability-distribution variant of LORE for the supervised setting, constraining the KL divergence between the fine-tuned model's output distributions and those of the pretrained model. This approach requires only class text embeddings, not sample labels. Applied to TeCoA, it yields a second regularized variant: TeCoA + KL.

Table 11: A comprehensive evaluation of clean and adversarial performance is conducted across various image classification datasets using the ViT-B/32 CLIP model. All models are trained on ImageNet and evaluated in a zero-shot setting across diverse benchmarks. Table demonstrates the increase (↑) in performance of our method relative to the corresponding FARE models.

| Eval. | Vision encoder | ImageNet | CalTech | Cars | CIFAR10 | CIFAR100 | DTD | EuroSAT | FGVC | Flowers | ImageNet-R | ImageNet-S | PCAM | OxfordPets | STL-10 | Average Zero-shot | |
|---|---|---|---|---|---|---|---|---|---|---|---|---|---|---|---|---|---|
| clean | CLIP | 59.8 | 84.1 | 59.6 | 89.7 | 63.3 | 44.4 | 46.1 | 19.6 | 66.3 | 69.3 | 42.3 | 62.3 | 87.5 | 97.2 | 64.0 | |
| | FARE$^1$ | 56.6 | 84.0 | 56.3 | 86.4 | 61.1 | 40.5 | 27.2 | 18.1 | 62.0 | 66.4 | 40.5 | 55.5 | 86.1 | 95.8 | 60.0 | |
| | LORE$^1$ | 57.4 | 84.4 | 55.9 | 88.5 | 64.5 | 40.1 | 29.9 | 16.7 | 61.3 | 67.2 | 41.5 | 53.8 | 86.9 | 96.3 | 60.5 | ↑0.5 |
| | FARE$^2$ | 52.9 | 82.2 | 49.7 | 76.3 | 51.1 | 36.4 | 18.4 | 15.7 | 53.3 | 60.4 | 35.9 | 48.2 | 82.7 | 93.0 | 54.1 | |
| | LORE$^2$ | 55.7 | 83.0 | 51.0 | 83.4 | 59.7 | 37.2 | 23.0 | 15.9 | 54.5 | 63.4 | 39.3 | 51.2 | 84.3 | 94.5 | 57.0 | ↑2.9 |
| | FARE$^4$ | 42.6 | 78.1 | 36.5 | 55.9 | 35.8 | 28.8 | 15.7 | 10.6 | 36.1 | 49.3 | 27.1 | 50.0 | 71.8 | 85.6 | 44.7 | |
| | LORE$^4$ | 50.1 | 80.3 | 40.1 | 72.4 | 49.6 | 32.4 | 17.7 | 11.4 | 39.7 | 55.1 | 33.6 | 50.0 | 79.3 | 90.4 | 50.2 | ↑5.5 |
| | FARE$^6$ | 33.0 | 73.0 | 24.7 | 40.0 | 24.9 | 23.7 | 15.2 | 6.24 | 22.7 | 39.5 | 20.2 | 50.0 | 52.4 | 75.0 | 36.0 | |
| | LORE$^6$ | 42.6 | 75.3 | 28.7 | 61.5 | 35.8 | 26.1 | 16.4 | 8.25 | 25.8 | 45.2 | 26.2 | 50.0 | 69.3 | 84.2 | 42.5 | ↑6.5 |
| | FARE$^8$ | 27.6 | 69.1 | 17.0 | 34.2 | 20.2 | 20.6 | 15.0 | 4.68 | 16.5 | 34.1 | 16.8 | 50.0 | 37.6 | 67.3 | 31.0 | |
| | LORE$^8$ | 41.3 | 74.6 | 24.9 | 61.5 | 35.9 | 24.5 | 16.0 | 7.14 | 22.8 | 43.0 | 24.4 | 50.0 | 67.4 | 83.5 | 41.2 | ↑10.2 |
| | FARE$^{10}$ | 23.2 | 66.0 | 12.8 | 31.3 | 17.5 | 18.8 | 15.0 | 4.11 | 13.7 | 30.2 | 14.4 | 50.0 | 27.6 | 61.9 | 28.0 | |
| | LORE$^{10}$ | 40.5 | 74.3 | 23.8 | 64.8 | 38.8 | 24.3 | 16.2 | 6.75 | 22.0 | 41.9 | 22.9 | 50.0 | 66.4 | 84.2 | 41.2 | ↑13.2 |
| $\varepsilon = 1.0$ | CLIP | 0.0 | 0.0 | 0.0 | 0.0 | 0.0 | 0.0 | 0.0 | 0.0 | 0.0 | 0.0 | 0.1 | 0.0 | 0.0 | 0.0 | 0.0 | |
| | FARE$^1$ | 27.8 | 68.6 | 16.1 | 61.0 | 35.6 | 22.5 | 6.1 | 2.9 | 30.6 | 34.4 | 22.5 | 24.7 | 55.8 | 82.2 | 35.6 | |
| | LORE$^1$ | 32.9 | 71.0 | 18.7 | 67.1 | 40.0 | 23.7 | 9.4 | 4.2 | 33.5 | 37.6 | 24.8 | 28.3 | 60.5 | 84.1 | 38.7 | ↑3.1 |
| | FARE$^2$ | 34.3 | 75.2 | 22.6 | 60.1 | 35.4 | 24.7 | 12.6 | 5.3 | 33.9 | 39.7 | 24.1 | 30.4 | 64.8 | 83.3 | 39.4 | |
| | LORE$^2$ | 39.3 | 76.3 | 23.3 | 67.0 | 43.2 | 26.4 | 12.3 | 6.5 | 35.8 | 42.4 | 26.4 | 39.0 | 68.5 | 85.6 | 42.5 | ↑3.1 |
| | FARE$^4$ | 33.2 | 74.8 | 21.4 | 44.9 | 28.0 | 22.4 | 14.0 | 5.8 | 27.3 | 37.1 | 21.3 | 50.2 | 59.3 | 77.7 | 37.2 | |
| | LORE$^4$ | 41.8 | 77.2 | 24.1 | 61.2 | 39.9 | 24.5 | 14.3 | 7.8 | 30.2 | 41.6 | 25.5 | 50.2 | 68.8 | 83.2 | 42.2 | ↑5.0 |
| | FARE$^6$ | 26.3 | 70.7 | 15.2 | 32.8 | 20.0 | 19.5 | 14.1 | 3.6 | 19.3 | 30.0 | 15.1 | 50.2 | 43.5 | 70.2 | 31.1 | |
| | LORE$^6$ | 36.2 | 74.4 | 18.9 | 52.2 | 30.6 | 20.8 | 15.4 | 6.3 | 22.3 | 34.2 | 22.1 | 50.2 | 60.5 | 78.1 | 37.4 | ↑6.3 |
| | FARE$^8$ | 23.1 | 66.9 | 10.8 | 28.2 | 16.7 | 17.7 | 14.6 | 3.1 | 15.5 | 25.3 | 12.5 | 50.2 | 30.9 | 62.7 | 27.3 | |
| | LORE$^8$ | 35.5 | 73.6 | 15.9 | 51.1 | 31.2 | 20.1 | 14.2 | 5.8 | 20.0 | 33.3 | 20.7 | 50.2 | 58.6 | 77.4 | 36.3 | ↑9.0 |
| | FARE$^{10}$ | 19.0 | 65.4 | 8.20 | 26.3 | 14.3 | 16.7 | 14.8 | 3.0 | 13.1 | 22.0 | 11.0 | 50.2 | 23.7 | 57.2 | 25.1 | |
| | LORE$^{10}$ | 33.8 | 71.2 | 14.3 | 51.7 | 30.1 | 19.5 | 12.9 | 4.2 | 18.9 | 31.5 | 19.4 | 50.2 | 53.9 | 76.7 | 35.0 | ↑9.9 |
| $\varepsilon = 2.0$ | CLIP | 0.0 | 0.0 | 0.0 | 0.0 | 0.0 | 0.0 | 0.0 | 0.0 | 0.0 | 0.0 | 0.1 | 0.0 | 0.0 | 0.0 | 0.0 | |
| | FARE$^1$ | 8.0 | 43.5 | 1.9 | 31.0 | 14.7 | 12.9 | 0.6 | 0.2 | 6.8 | 13.4 | 11.7 | 14.1 | 15.9 | 54.9 | 17.0 | |
| | LORE$^1$ | 13.1 | 49.0 | 3.3 | 37.9 | 19.0 | 14.2 | 2.5 | 0.5 | 10.1 | 17.6 | 13.1 | 19.1 | 23.1 | 61.2 | 20.8 | ↑3.8 |
| | FARE$^2$ | 19.3 | 59.9 | 7.7 | 41.2 | 22.8 | 17.8 | 9.6 | 1.5 | 16.4 | 24.2 | 15.9 | 23.4 | 38.6 | 68.6 | 26.7 | |
| | LORE$^2$ | 24.0 | 63.3 | 8.6 | 47.2 | 27.2 | 18.2 | 10.6 | 1.7 | 18.5 | 26.0 | 18.4 | 28.0 | 44.4 | 73.1 | 29.6 | ↑2.9 |
| | FARE$^4$ | 24.1 | 65.5 | 10.4 | 36.0 | 21.6 | 18.8 | 12.3 | 2.7 | 17.9 | 27.7 | 15.8 | 50.0 | 44.4 | 68.8 | 30.1 | |
| | LORE$^4$ | 32.6 | 69.5 | 12.4 | 50.8 | 29.6 | 20.9 | 13.0 | 3.3 | 21.6 | 32.3 | 20.0 | 50.1 | 55.9 | 76.1 | 35.0 | ↑4.9 |
| | FARE$^6$ | 20.2 | 64.6 | 8.4 | 27.4 | 16.7 | 17.1 | 13.0 | 1.8 | 14.3 | 23.6 | 11.8 | 50.2 | 33.3 | 62.1 | 26.5 | |
| | LORE$^6$ | 30.1 | 68.3 | 10.7 | 44.0 | 25.6 | 18.5 | 13.8 | 3.7 | 16.8 | 27.6 | 17.0 | 50.2 | 49.7 | 71.2 | 32.1 | ↑5.6 |
| | FARE$^8$ | 17.4 | 62.7 | 6.4 | 24.5 | 13.7 | 15.4 | 13.1 | 1.5 | 11.2 | 19.7 | 10.7 | 50.2 | 24.0 | 56.2 | 23.8 | |
| | LORE$^8$ | 30.9 | 68.8 | 10.5 | 43.3 | 25.7 | 18.5 | 13.5 | 3.2 | 16.6 | 27.8 | 17.0 | 50.2 | 49.2 | 71.6 | 31.9 | ↑8.1 |
| | FARE$^{10}$ | 15.1 | 60.0 | 5.0 | 23.5 | 11.8 | 14.3 | 13.5 | 1.7 | 10.6 | 18.5 | 9.2 | 50.2 | 18.4 | 52.5 | 22.2 | |
| | LORE$^{10}$ | 29.7 | 66.8 | 8.8 | 38.8 | 24.1 | 17.9 | 12.4 | 2.5 | 14.9 | 27.0 | 16.2 | 50.2 | 45.5 | 69.1 | 30.3 | ↑8.1 |
| $\varepsilon = 4.0$ | CLIP | 0.0 | 0.0 | 0.0 | 0.0 | 0.0 | 0.0 | 0.0 | 0.0 | 0.0 | 0.0 | 0.1 | 0.0 | 0.0 | 0.0 | 0.0 | |
| | FARE$^1$ | 0.3 | 6.3 | 0.0 | 1.7 | 2.0 | 2.3 | 0.0 | 0.0 | 0.1 | 2.6 | 2.4 | 0.9 | 0.0 | 5.3 | 1.8 | |
| | LORE$^1$ | 0.7 | 9.7 | 0.0 | 3.5 | 3.1 | 4.0 | 0.0 | 0.0 | 0.2 | 3.8 | 2.8 | 2.7 | 0.0 | 9.4 | 3.0 | ↑1.2 |
| | FARE$^2$ | 3.2 | 27.5 | 0.5 | 12.3 | 7.0 | 7.7 | 4.3 | 0.0 | 2.4 | 6.8 | 5.1 | 15.8 | 3.0 | 30.1 | 9.4 | |
| | LORE$^2$ | 5.7 | 31.1 | 0.7 | 13.0 | 8.2 | 9.7 | 0.8 | 0.0 | 3.1 | 8.3 | 6.5 | 18.2 | 7.2 | 33.5 | 10.8 | ↑1.4 |
| | FARE$^4$ | 10.7 | 46.3 | 1.5 | 19.7 | 11.8 | 11.9 | 10.2 | 0.6 | 6.4 | 11.4 | 8.7 | 45.2 | 16.2 | 46.1 | 18.2 | |
| | LORE$^4$ | 17.8 | 54.2 | 2.8 | 27.4 | 16.8 | 14.4 | 10.0 | 0.6 | 8.0 | 16.4 | 11.7 | 48.4 | 25.5 | 56.1 | 22.5 | ↑4.3 |
| | FARE$^6$ | 11.6 | 50.5 | 1.6 | 19.2 | 9.8 | 12.1 | 11.1 | 0.6 | 6.3 | 12.7 | 7.4 | 50.2 | 15.8 | 46.0 | 18.7 | |
| | LORE$^6$ | 19.2 | 57.0 | 3.5 | 26.2 | 16.4 | 13.9 | 12.7 | 1.0 | 8.9 | 16.9 | 10.5 | 50.2 | 26.9 | 57.0 | 23.2 | ↑4.5 |
| | FARE$^8$ | 10.9 | 50.0 | 1.5 | 18.3 | 9.2 | 11.4 | 11.8 | 0.7 | 6.3 | 11.9 | 6.5 | 50.2 | 12.4 | 44.3 | 18.0 | |
| | LORE$^8$ | 21.7 | 58.8 | 4.1 | 28.0 | 17.2 | 13.8 | 12.8 | 1.1 | 9.5 | 17.7 | 10.9 | 50.2 | 31.5 | 59.0 | 24.2 | ↑6.2 |
| | FARE$^{10}$ | 9.03 | 48.3 | 1.1 | 17.7 | 8.3 | 10.4 | 11.5 | 0.4 | 5.5 | 11.1 | 5.4 | 50.2 | 10.6 | 41.9 | 17.1 | |
| | LORE$^{10}$ | 21.1 | 56.8 | 3.5 | 22.1 | 14,8 | 13.7 | 11.8 | 0.9 | 9.3 | 17.4 | 10.6 | 50.2 | 28.8 | 52.9 | 22.5 | ↑5.4 |

For classification using probability distributions induced by the contrastive model, we formulate the constrained optimization problem:

$$\min_{\theta \in \Theta} E_{x,y \sim D}\big[L_{\sup}(x, y; \theta)\big] \quad \text{s.t.} \quad D_{\mathrm{KL}}(p_{\theta_0}(\cdot \mid x), p_\theta(\cdot \mid x)) \le \rho \cdot m(x), \quad \text{where} \quad m(x) = H(p_{\theta_0}(\cdot \mid x))$$

Table 12: **Evaluation under high-intensity adversarial attacks.** A comprehensive assessment of clean and adversarial performance is conducted across various image classification datasets using the ViT-B/32 CLIP model. All models are trained on ImageNet and evaluated in a zero-shot setting across diverse benchmarks.

| Eval. | Vision encoder | ImageNet | CalTech | Cars | CIFAR10 | CIFAR100 | DTD | EuroSAT | FGVC | Flowers | ImageNet-R | ImageNet-S | PCAM | OxfordPets | STL-10 | Average Zero-shot | |
|---|---|---|---|---|---|---|---|---|---|---|---|---|---|---|---|---|---|
| $\varepsilon = 6.0$ | FARE[6] | 5.5 | 28.5 | 0.1 | 11.4 | 6.2 | 7.7 | 10.4 | 0.0 | 2.7 | 5.8 | 3.5 | 50.2 | 3.6 | 30.4 | 12.3 | |
| | LORE[6] | 10.4 | 40.1 | 0.6 | 13.7 | 9.6 | 9.9 | 11.9 | 0.1 | 4.5 | 9.4 | 6.6 | 50.2 | 11.4 | 40.1 | 16.0 | ↑3.7 |
| | FARE[8] | 5.5 | 30.6 | 0.0 | 12.5 | 6.1 | 7.8 | 11.4 | 0.0 | 3.1 | 6.2 | 3.7 | 50.2 | 3.9 | 29.6 | 12.7 | |
| | LORE[8] | 12.9 | 44.6 | 1.4 | 15.0 | 10.9 | 10.4 | 12.2 | 0.6 | 5.3 | 11.0 | 7.4 | 50.2 | 15.3 | 43.2 | 17.5 | ↑4.8 |
| | FARE[10] | 5.3 | 31.4 | 0.0 | 13.7 | 5.4 | 7.3 | 11.8 | 0.0 | 3.2 | 6.1 | 3.6 | 50.2 | 4.2 | 28.5 | 12.7 | |
| | LORE[10] | 13.6 | 45.0 | 1.5 | 10.7 | 9.6 | 10.1 | 11.3 | 0.6 | 5.4 | 11.3 | 7.1 | 50.2 | 14.2 | 35.3 | 16.3 | ↑3.6 |
| $\varepsilon = 8.0$ | FARE[6] | 1.9 | 16.8 | 0.0 | 5.9 | 3.5 | 5.7 | 9.2 | 0.0 | 0.7 | 3.2 | 2.2 | 50.2 | 0.7 | 13.8 | 8.6 | |
| | LORE[6] | 4.8 | 24.3 | 0.2 | 6.7 | 5.0 | 7.4 | 8.1 | 0.0 | 2.0 | 5.5 | 4.0 | 50.2 | 3.6 | 22.3 | 10.7 | ↑2.1 |
| | FARE[8] | 2.2 | 19.1 | 0.0 | 8.4 | 3.8 | 5.4 | 10.0 | 0.0 | 1.4 | 3.2 | 2.2 | 50.2 | 0.9 | 16.2 | 9.3 | |
| | LORE[8] | 7.5 | 30.4 | 0.4 | 7.6 | 6.2 | 8.0 | 10.2 | 0.0 | 3.8 | 6.5 | 4.8 | 50.2 | 6.6 | 25.6 | 12.3 | ↑3.0 |
| | FARE[10] | 2.6 | 19.5 | 0.0 | 9.1 | 4.0 | 5.2 | 10.1 | 0.0 | 1.6 | 3.3 | 2.1 | 50.2 | 1.2 | 16.7 | 9.4 | |
| | LORE[10] | 8.0 | 31.9 | 0.4 | 5.0 | 5.3 | 8.0 | 10.6 | 0.0 | 4.0 | 6.8 | 5.0 | 50.2 | 6.8 | 20.0 | 11.8 | ↑2.4 |
| $\varepsilon = 10.0$ | FARE[6] | 0.7 | 9.2 | 0.0 | 2.8 | 1.9 | 3.3 | 6.8 | 0.0 | 0.1 | 1.5 | 1.2 | 50.0 | 0.2 | 4.4 | 6.3 | |
| | LORE[6] | 1.7 | 13.9 | 0.0 | 2.8 | 2.7 | 4.8 | 0.1 | 0.0 | 0.7 | 3.2 | 2.2 | 50.1 | 0.4 | 9.5 | 6.9 | ↑0.6 |
| | FARE[8] | 0.8 | 10.5 | 0.0 | 3.9 | 2.2 | 3.6 | 8.0 | 0.0 | 0.5 | 1.5 | 1.1 | 50.1 | 0.5 | 6.5 | 6.8 | |
| | LORE[8] | 3.2 | 18.9 | 0.1 | 3.7 | 3.3 | 5.6 | 2.9 | 0.0 | 1.6 | 4.2 | 3.0 | 50.1 | 2.1 | 13.8 | 8.4 | ↑1.6 |
| | FARE[10] | 1.0 | 11.5 | 0.0 | 5.1 | 2.3 | 3.4 | 8.5 | 0.0 | 0.6 | 1.8 | 1.2 | 50.1 | 0.6 | 7.6 | 7.1 | |
| | LORE[10] | 4.0 | 20.2 | 0.1 | 2.1 | 3.2 | 6.0 | 7.1 | 0.0 | 1.7 | 4.9 | 3.3 | 50.2 | 2.8 | 11.2 | 8.7 | ↑1.6 |

Table 13: Comparison of Adaptive Margin $m(x)$ vs. Fixed Margins (ViT-B/32 on ImageNet-100). The Avg Robust Acc column is the average of the three adversarial perturbation levels ($\varepsilon = 1, 2, 4$).

| Train $\varepsilon$ | Method | $\rho$ | Clean | $\varepsilon = 1$ | $\varepsilon = 2$ | $\varepsilon = 4$ | Avg Robust Acc |
|---|---|---|---|---|---|---|---|
| 2 | LORE[2]$_{m(x)}$ | 0.1 | 68.81 | **51.51** | **33.71** | 9.93 | **31.72** |
| | LORE[2] | 0.6 | 70.12 | 24.32 | 18.65 | 0.56 | 14.51 |
| | LORE[2] | 0.8 | 69.65 | 45.65 | 29.45 | 5.65 | 26.92 |
| | LORE[2] | 1.5 | 63.45 | 41.23 | 25.47 | 3.64 | 23.45 |
| | FARE[2] | — | 62.23 | 40.12 | 23.15 | 3.40 | 22.22 |
| 4 | LORE[4]$_{m(x)}$ | 0.1 | 66.91 | **46.99** | **35.71** | 17.58 | **33.43** |
| | LORE[4] | 0.6 | 67.21 | 19.50 | 7.21 | 1.23 | 9.31 |
| | LORE[4] | 0.8 | 64.32 | 40.21 | 32.23 | 14.68 | 29.04 |
| | LORE[4] | 1.5 | 55.24 | 33.21 | 29.32 | 11.74 | 24.76 |
| | FARE[4] | — | 53.68 | 34.85 | 28.78 | 12.87 | 25.50 |

Building on this formulation, LORE's final objective function can be computed as before. The framework adjusts the proximity constraint by model confidence: low entropy (high confidence) tightens the constraint, while high entropy relaxes it for greater adaptation flexibility.

**Results.** The experimental setting involved fine-tuning ViT-B/32 CLIP models for 15 epochs on ImageNet-100 using a learning rate of $1e - 5$. The proximity constraint hyperparameter used $\rho = 0.1$. The results are shown in Table 14.

TeCoA consistently benefits from both LORE-based ($l_2$) and KL regularization, achieving improved robustness while maintaining clean performance. Notably, for high adversarial training budgets (e.g., Train $\varepsilon = 8$), the TeCoA + $l_2$ method yields a massive increase in clean accuracy compared to the baseline TeCoA. These results demonstrate that proximity-based constraints—whether applied in embedding space ($l_2$) or probability space (KL)—can effectively stabilize adversarial fine-tuning in supervised frameworks as well.

Table 14: Extension to Supervised Fine-Tuning (ViT-B/32 CLIP on ImageNet-100). The Avg Robust Acc column is the average of the four adversarial perturbation levels ($\varepsilon = 1, 2, 4, 8$).

| Train $\varepsilon$ | Method | Clean | $\varepsilon = 1$ | $\varepsilon = 2$ | $\varepsilon = 4$ | $\varepsilon = 8$ | Avg Robust Acc |
|---|---|---|---|---|---|---|---|
| | TeCoA$^2$ | 60.04 (ref) | 49.11 | 35.94 | 14.73 | 0.50 | 25.07 (ref) |
| 2 | **TeCoA$^2$ + $l_2$** | **75.97 (+15.93 ↑)** | 53.24 | 39.12 | 18.12 | 1.23 | **27.93 (+2.86 ↑)** |
| | TeCoA$^2$ + KL | 66.50 (+6.46 ↑) | 53.04 | 39.53 | 16.94 | 0.60 | 27.53 (+2.46 ↑) |
| | TeCoA$^4$ | 49.55 (ref) | 44.20 | 38.21 | 19.80 | 1.03 | 25.81 (ref) |
| 4 | **TeCoA$^4$ + $l_2$** | **73.12 (+23.57 ↑)** | 56.20 | 53.12 | 49.12 | 5.23 | **40.92 (+15.11 ↑)** |
| | TeCoA$^4$ + KL | 56.01 (+6.46 ↑) | 48.62 | 42.03 | 22.18 | 1.25 | 28.52 (+2.71 ↑) |
| | TeCoA$^8$ | 30.22 (ref) | 24.72 | 19.09 | 7.41 | 2.35 | 13.39 (ref) |
| 8 | **TeCoA$^8$ + $l_2$** | **67.23 (+37.01 ↑)** | 36.21 | 26.40 | 15.12 | 4.23 | **20.49 (+7.10 ↑)** |
| | TeCoA$^8$ + KL | 38.06 (+7.84 ↑) | 27.69 | 21.94 | 8.75 | 2.82 | 15.30 (+1.91 ↑) |

## J.8 Validation on Downstream Tasks (Segmentation)

To confirm LORE's generalizability beyond classification, we evaluated its effectiveness on semantic segmentation, a critical VLM downstream task. We use the open-source benchmark from Kowalczuk et al. [2024] to have a fair comparison with our baselines.

We assessed the robustness of the DINOv2 ViT-S/14 model on the ADE20k segmentation dataset. The results in Table 15 show that the LORE-hardened model improves robustness against adversarial attacks in the embedding space while maintaining a favorable clean accuracy trade-off.

Table 15: Segmentation Results on ADE20k (DINOv2 ViT-S/14). LORE[8] and FARE[8] denote models trained with an $\varepsilon = 8/255$ perturbation budget. mIoU is reported for clean data and under an adversarial attack in the embedding space.

| Method | Clean mIoU | Embed Attack mIoU |
|---|---|---|
| Pretrained | 0.42 | 0.23 |
| LORE[8] | **0.43** | **0.35** |
| FARE[8] | 0.23 | 0.15 |

## K Computation and Efficiency Analysis

In this section, we analyze the computational aspects of LORE in terms of training time, efficiency, and convergence behavior. While LORE introduces an additional dual network and constraint enforcement mechanism, we find that its cost remains practical and comparable to FARE baselines.

### K.1 Convergence Efficiency.

To compare the training efficiency of FARE and LORE, we measure the total training time (in minutes) required to reach specific robust accuracy (RAcc) thresholds on the validation set. Table 16 reports this comparison across various model backbones, including ViT-B/16, ViT-B/32, ConvNeXt-B, ViT-S/14, and ViT-B/14. LORE often reaches target RAcc levels in fewer training minutes than FARE, highlighting its superior optimization efficiency and stability.

### K.2 Impact of $\lambda_\omega$ architecture on Training Time.

The results in Table 17 show that LORE's current dual network design is not only significantly more efficient than using a separate pretrained CLIP model as a dual network, but also achieves comparable runtime to simpler parameterizations (scalar or linear forms). This demonstrates that LORE achieves computational efficiency without sacrificing expressive capacity.

Table 16: Training time (in minutes) required to reach different robust accuracy (RAcc) thresholds under PGD attack ($\varepsilon = 2/255$). Lower is better. All models were trained using 4 NVIDIA H100 80GB GPUs.

| Model | Method | 10% | 20% | 30% | 35% | Dataset |
|-------|--------|-----|-----|-----|-----|---------|
| ViT-B/16 | FARE | 23 | 38 | 64 | 98 | ImageNet-100 |
|          | LORE | 11 | 17 | 23 | 31 | |
| ViT-B/32 | FARE | 30 | 86 | – | – | ImageNet |
|          | LORE | 25 | 68 | 185 | – | |
| ViT-B/32 | FARE | 62 | 115 | 165 | 285 | ImageNet-100 |
|          | LORE | 47 | 84 | 148 | 268 | |
| ConvNeXt-B | FARE | 19 | 26 | 34 | 40 | ImageNet-100 |
|            | LORE | 52 | 57 | 60 | 63 | |
| ViT-S/14 | FARE | 25 | 43 | 100 | 181 | ImageNet |
|          | LORE | 39 | 71 | 124 | 126 | |
| ViT-B/14 | FARE | 22 | 28 | 43 | 56 | ImageNet |
|          | LORE | 29 | 32 | 42 | 53 | |

Table 17: Training time (in seconds) for 50 iterations of LORE using 4×H100 GPUs across different architectures for $\lambda_\omega$. The *Linear* model uses a single-layer projection: `nn.Linear(self.embedding_size, 1)`. Lower is better.

| Architecture | Training Time (s) |
|--------------|-------------------|
| LORE (Current Dual Network) | 551 |
| Scalar $\lambda$ (input-independent) | 533 |
| Linear $\lambda$ (input-independent) | 540 |
| Pretrained CLIP | 692 |

## K.3 Impact of $K$ on Training Time.

As described in Section 5, LORE alternates between $K$ primal updates and one dual update per batch. While Section I analyzes the impact of $K$ on final performance, here in Fig. 15, we empirically examine how varying $K$ affects training time.

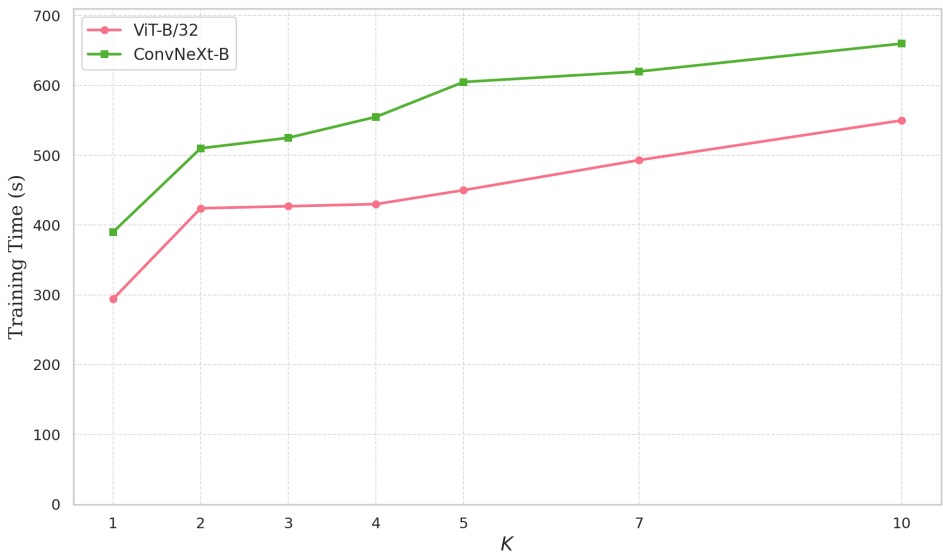

Figure 15: Figure: Training time (in seconds) for completing 30 training iterations of LORE under different values of $K$, using 8×H100 GPUs. Results are reported for two architectures: ViT-B/32 and ConvNeXt-B. Increasing $K$ slightly raises the training time due to more frequent primal updates, with consistent trends across both models.

