# OpenReview forum: "LORE: Lagrangian-Optimized Robust Embeddings for Visual Encoders"
_NeurIPS.cc/2025/Conference — NeurIPS 2025 poster_

### Official Review · Reviewer_Jf2T · 2025-06-17

**Clarity:** 2
**Significance:** 2
**Originality:** 3
**Rating:** 3
**Confidence:** 5

**Summary:**

This paper proposes Lagrangian-Optimized Robust Embeddings (LORE), a novel unsupervised adversarial fine-tuning framework, which enhances the robustness of visual encoders while preserving nominal performance without relying on heuristic loss weighting. Extensive experiments show that LORE significantly improves zero-shot adversarial robustness with minimal degradation in clean data accuracy and outperforms FARE consistently. However, the paper still needs further refinement and revision. Further experimental validation and analysis are recommended to be added.

**Questions:**

Some problems are:
1、In this paper, FARE is considered as a baseline for evaluating the proposed approach. Hence, it is important to highlight the contributions of the proposed method LORE compared with FARE in the introduction.

2、In Section 2 Related Work, there are recent studies that highlight the vulnerability of VLMs to adversarial attacks. However, the corresponding description of the relevant references is just too limited to fit the regular structure of a paper.

3、For Figure 1(a), the meaning of varying `\rho` is not mentioned in the paper. Further explanation is necessary to be added to the manuscript.

4、To empirically validate the approach, the majority of our experiments were conducted and all models were trained for 2 epochs on ImageNet or 5 epochs on ImageNet-100. Does this signify that the structure of the model is very simple and therefore the training process is comparatively fast?

5、A comprehensive evaluation of clean and adversarial performance is conducted across various image classification datasets in Table 3. But there are experimental results of ‘0’ and no experimental results for `\epsilon`=8. Further experimental validations are necessary to be added to the manuscript.

**Ethical Concerns:**

["NO or VERY MINOR ethics concerns only"]

**Final Justification:**

I have read the authors' response. The authors have addressed some of my concerns. However, a major revision is required. In particular, more experiments should be given. Therefore, I will keep my original rating.

**Limitations:**

Mentioned in the paper

**Paper Formatting Concerns:**

No.

**Quality:**

2

**Strengths And Weaknesses:**

Pros: Extensive experiments and Interesting Topic.

Cons:  Further validation and analysis are recommended to be added.

---

> ### Author Rebuttal · Authors · 2025-07-31
>
> # Response to Reviewer Jf2T
>
> We appreciate that Reviewer Jf2T finds our method novel, our experimental results significant, and the topic interesting.
>
> # Response to the questions
>
> > Answer for Q1
>
> Thank you for this valuable suggestion. While we have already cited the FARE paper in the introduction (Line 41), we agree that a clearer comparison between LORE and FARE at the outset would better contextualize our contribution. In the revised version, we will explicitly highlight the key distinctions and motivations for proposing LORE in contrast to FARE within the introduction.
>
> > Answer for Q2
>
> In the initial version of the paper, due to space limitations, we kept the Related Work section concise and focused on maintaining a coherent narrative rather than elaborating on each individual reference. Additionally, we would like to clarify that our primary focus is on improving the robustness of visual encoders, rather than VLMs as a whole. While we acknowledge the growing body of work on adversarial robustness in VLMs, we have deliberately centered our discussion on methods most relevant to image encoder fine-tuning.
>
> That said, we appreciate the suggestion and in the camera-ready version, where we are allowed an additional page, we plan to expand this section and provide deeper analysis and categorization of relevant literature, particularly those highlighting VLM vulnerabilities and defenses.
>
> > Answer for Q3
>
> Thank you for your comment. To clarify, Figure 1(a) is not related to varying the hyperparameter $\rho$, $\rho=0.1$ was used for this experiment. If the comment is instead referring to Figure 1(b), then as explained in Section 6.1, $\rho$ defines the threshold for the proximity constraint (see Eq. 5) and directly controls the trade-off between nominal (clean) accuracy and adversarial robustness. Varying $\rho$ adjusts the allowed deviation from the reference encoder, enabling a principled way to navigate this trade-off.
>
> We included Figure 1(b) in Section 4 to motivate the LORE framework by showcasing its empirical advantage over naïve regularization methods, even before formally introducing the full formulation. Additionally, we explicitly reference and explain the meaning of varying $\rho$ in Lines 229–231, where we describe how LORE achieves controllability through this parameter. The theoretical and empirical roles of $\rho$ are then discussed in full detail in Section 6.1 and are further supported by our robustness suboptimality analysis.
>
> We hope this clears up the confusion, and we are happy to provide further clarification if needed.
>
> > Answer for Q4
>
> The models we fine-tune, such as CLIP ViT-B/32, DINOv2, and ConvNeXt-B, are large, pre-trained foundation models, not simple architectures. The relatively low number of fine-tuning epochs (2 for ImageNet and 5 for ImageNet-100) is not due to model simplicity; it signifies the efficiency of this approach for fine-tuning on large-scale datasets.
>
> The number of epochs was selected based on training convergence behavior. Specifically, for the large-scale ImageNet dataset, 2 epochs were sufficient to reach strong convergence, **consistent with the FARE paper**, and yielded robust performance in both in-domain and zero-shot evaluations. Similarly, given the smaller size and diversity of ImageNet-100, we found that 5 epochs were adequate. Additionally, we provide training curves for several experiments in Appendix G.2 (e.g., Figure 10), where the model's convergence can be clearly observed.
>
> > Answer for Q5
>
> Thank you for your comment. The '0' values for the original CLIP model (rows labeled "CLIP") under adversarial attacks in Table 3 are expected and reflect the severe vulnerability of the base model. This underscores the motivation for robust fine-tuning methods like LORE, as CLIP without adversarial training exhibits negligible robustness. As shown in the table, LORE consistently outperforms FARE on average across different settings, further validating its effectiveness. Regarding other '0' values in Table 3 (e.g., under evaluation at $\epsilon=4$), these correspond to models that were trained with lower perturbation strengths (e.g., $\epsilon=1,2$ for FARE1 and LORE2). As expected, models trained with smaller perturbations typically do not generalize well to stronger attacks—highlighting a common limitation of fixed-$\epsilon$ training.
>
> As for high-intensity perturbations (e.g., $\epsilon=8$), we have indeed conducted those evaluations. The corresponding results are reported in Tables 9 and 10 of Appendix J.4. Due to space constraints in the main manuscript, these extended results are included in the appendix, as noted in Line 188 of the Results section.
>
> Regarding further experimental validations, in addition to the results presented in Section 6, we conducted additional evaluations in **Appendix J**, including Square Attack, a black-box adversarial attack, and evaluation under Gaussian noise corruption.
>
> # Further experimental validation
>
> ## 1. Validating our choice of sample-specific margins.
>
> To validate the design choice between sample-specific and sample-invariant tolerance settings, experiments were conducted using ViT-B/32 CLIP models. These models were fine-tuned for 5 epochs on ImageNet-100. For sample-invariant settings, $\rho=0.8$ was empirically chosen as the best-performing fixed tolerance among tested candidates. The baseline for comparison was a prior LORE setup with $m(x)=\|\phi_{\theta_0}(x)\|^2_2$ and $\rho=0.1$, which represents the sample-specific adaptive tolerance.
>
> The table below presents the results:
>
> | Method                   | ρ   | Clean                   | $\epsilon=1$ | $\epsilon=2$ | $\epsilon=4$ | Avg Robust Acc            |
> |--------------------------|-----|-------------------------|--------------|--------------|--------------|---------------------------|
> | LORE$^2$ + $m(x)$        | 0.1 | 68.81 (ref)             | 51.51        | 33.71        | 9.93         | 31.72 (ref)               |
> | LORE$^2$                 | 0.6 | 70.12 (+1.31 ↑)         | 24.32        | 18.65        | 0.56         | 14.51 (−17.21 ↓)          |
> | LORE$^2$                 | 0.8 | 69.65 (+0.84 ↑)         | 45.65        | 29.45        | 5.65         | 26.92 (−4.80 ↓)           |
> | LORE$^2$                 | 1.5 | 63.45 (−5.36 ↓)         | 41.23        | 25.47        | 3.64         | 23.45 (−8.27 ↓)           |
> | FARE$^2$                 | —   | 62.23 (−6.58 ↓)         | 40.12        | 23.15        | 3.40         | 22.22 (−9.50 ↓)           |
> |---|---|---|---|---|---|---|
> | LORE$^4$ + $m(x)$        | 0.1 | 66.91 (ref)             | 46.99        | 35.71        | 17.58        | 33.43 (ref)               |
> | LORE$^4$                 | 0.6 | 67.21 (+0.30 ↑)         | 19.50        | 7.21         | 1.23         | 9.31 (−24.12 ↓)           |
> | LORE$^4$                 | 0.8 | 64.32 (−2.59 ↓)         | 40.21        | 32.23        | 14.68        | 29.04 (−4.39 ↓)           |
> | LORE$^4$                 | 1.5 | 55.24 (−11.67 ↓)        | 33.21        | 29.32        | 11.74        | 24.76 (−8.67 ↓)           |
> | FARE$^4$                 | —   | 53.68 (−13.23 ↓)        | 34.85        | 28.78        | 12.87        | 25.50 (−7.93 ↓)           |
>
> Fixed tolerances show limitations: 0.6 is too restrictive, hurting robustness, while 1.5 is too permissive, mimicking FARE. Finding a universally optimal fixed tolerance is difficult. The sample-specific margin (e.g., LORE$^2$+$m(x)$ and LORE$^4$+$m(x)$ with $\rho=0.1$) is superior, adapting to embedding scale, consistently offering better robustness across various perturbations and training conditions while maintaining clean accuracy and stability.
>
> ## 2. Extentions to supervised finetuning
>
> We also demonstrate that LORE's proximity constraints can be extended to supervised adversarial fine-tuning, applying it to TeCoA [1]. We introduce two variants: TeCoA + $l_2$, which constrains the $l_2$ distance between fine-tuned and pretrained image embeddings, and TeCoA + KL, which uses KL divergence to constrain output probability distributions. Experiments with ViT-B/32 CLIP models on ImageNet-100 demonstrate that both regularized methods consistently enhance robustness and clean accuracy compared to baseline TeCoA, across various perturbation strengths. This proves LORE's broad applicability and the stabilizing effect of proximity-based constraints in both embedding and probability spaces for supervised adversarial fine-tuning. For more details, refer to our comment to Reviewer CYtr.
>
>
> ---
>
> [1] Mao, C., Geng, S., Yang, J., Wang, X. E., and Vondrick, C. Understanding zero-shot adversarial robustness for large-scale models. In ICLR, 2023.

---

### Official Review · Reviewer_TnAL · 2025-06-28

**Clarity:** 3
**Significance:** 2
**Originality:** 2
**Rating:** 4
**Confidence:** 2

**Summary:**

In this paper, the authors propose a Lagrangian-based optimization method to train a robust visual encoder, i.e., CLIP and DINOv2. Compared to the previous method, their method is more stable during the training process and achieves a better tradeoff between clean and adversarial accuracy, which is very important for adversarial robustness. Their experiments show that their method achieves higher adversarial accuracy on the image classification problem while the clean accuracy does not drop significantly.

**Questions:**

In your method, you use a sample-specific tolerance margin, which is related to the norm of the original feature embedding. What would the results be if we used a sample-invariant tolerance?

**Ethical Concerns:**

["NO or VERY MINOR ethics concerns only"]

**Final Justification:**

After the discussion with the authors, I felt most of my concerns were addressed. They added some results on the segmentation task to show that the method can improve the robustness of CLIP on many downstream tasks. They highlighted their novelty compared to prior work. Although I still think the innovation of using a trainable $\lambda$ is somewhat limited, I raised my rate given the improved performance and their detailed response.

**Limitations:**

Yes, they discuss their limitations in the last section.

**Paper Formatting Concerns:**

The paper is well-written.

**Quality:**

2

**Strengths And Weaknesses:**

Pros:
* In Section 4, they show that simply training the CLIP using an adversarial loss $L_{FARE}$ and a fixed-weighted regularizer could not solve the problem, which provides a good motivation for their method.
* The paper is well-written with clear logic and results.
* Their method is effective, achieving better robustness under adversarial attack while having minimal degradation on the clean accuracy.
* They validate the efficiency of their method on various dataset sizes and adversarial attack budgets, indicating the robustness of their method over general settings.

Cons:
* In the experiment part, the authors use the image classification task as the measurement to verify the robustness. Since the CLIP representation has been used in various applications such as detection and VLM, could the authors validate the robustness on those tasks?
* There are many other related methods proposed to improve the robustness of visual encoders [1,2], so I suggest you compare your method with more baselines rather than only comparing with FARE.
* The novelty of this work is limited. There has already been the unconstrained version of the adversarial optimization problem. And the difference between the proposed loss function Eq. (6) and the regularized loss function Eq. (3) only lies in the trainable weight $\lambda(x)$.
* In Line 130, the embedding space is denoted as $\mathcal{Y}$, while $y$ is used to represent the label of a data point $x$. It would be better to distinguish these two.

[1]: Robust CLIP: Unsupervised Adversarial Fine-Tuning of Vision Embeddings for Robust Large Vision-Language Models.
[2]: Improving robustness of CLIP by adversarial training enhanced by brain activity.

---

> ### Author Rebuttal · Authors · 2025-07-31
>
> We appreciate that Reviewer TnAL appreciates the superior trade-offs and stability of our method, and finds it important for adversarial robustness.
>
> # Response to Questions
>
> > **In your method, you use a sample-specific tolerance margin, which is related to the norm of the original feature embedding. What would the results be if we used a sample-invariant tolerance?**
>
> Thank you for the insightful question. Using a sample-invariant tolerance instead of a sample-specific margin indeed offers an alternative, but it is challenging because it requires extensive, model-dependent hyperparameter tuning that may not generalize. Therefore, the authors chose a sample-specific margin using $m(x)=\|\phi_{\theta_0}(x)\|^2_2$, which provides an adaptive, scale-invariant tolerance tied to the pre-trained embedding's norm, reducing manual tuning and preserving semantic structure. additionally this relates the constraints geometrically to the cosine similarity of embeddings.
>
> ## Experimental Comparison
>
> To validate the design choice between sample-specific and sample-invariant tolerance settings, experiments were conducted using ViT-B/32 CLIP models. These models were fine-tuned for 5 epochs on ImageNet-100. For sample-invariant settings, $\rho=0.8$ was empirically chosen as the best performing fixed tolerance among tested candidates. The baseline for comparison was a prior LORE setup with $m(x)=\|\phi_{\theta_0}(x)\|^2_2$ and $\rho=0.1$, which represents the sample-specific adaptive tolerance.
>
> The table below presents the results:
>
> | Method                   | ρ   | Clean                   | $\epsilon=1$ | $\epsilon=2$ | $\epsilon=4$ | Avg Robust Acc            |
> |--------------------------|-----|-------------------------|--------------|--------------|--------------|---------------------------|
> | LORE$^2$ + $m(x)$        | 0.1 | 68.81 (ref)             | 51.51        | 33.71        | 9.93         | 31.72 (ref)               |
> | LORE$^2$                 | 0.6 | 70.12 (+1.31 ↑)         | 24.32        | 18.65        | 0.56         | 14.51 (−17.21 ↓)          |
> | LORE$^2$                 | 0.8 | 69.65 (+0.84 ↑)         | 45.65        | 29.45        | 5.65         | 26.92 (−4.80 ↓)           |
> | LORE$^2$                 | 1.5 | 63.45 (−5.36 ↓)         | 41.23        | 25.47        | 3.64         | 23.45 (−8.27 ↓)           |
> | FARE$^2$                 | —   | 62.23 (−6.58 ↓)         | 40.12        | 23.15        | 3.40         | 22.22 (−9.50 ↓)           |
> |---|---|---|---|---|---|---|
> | LORE$^4$ + $m(x)$        | 0.1 | 66.91 (ref)             | 46.99        | 35.71        | 17.58        | 33.43 (ref)               |
> | LORE$^4$                 | 0.6 | 67.21 (+0.30 ↑)         | 19.50        | 7.21         | 1.23         | 9.31 (−24.12 ↓)           |
> | LORE$^4$                 | 0.8 | 64.32 (−2.59 ↓)         | 40.21        | 32.23        | 14.68        | 29.04 (−4.39 ↓)           |
> | LORE$^4$                 | 1.5 | 55.24 (−11.67 ↓)        | 33.21        | 29.32        | 11.74        | 24.76 (−8.67 ↓)           |
> | FARE$^4$                 | —   | 53.68 (−13.23 ↓)        | 34.85        | 28.78        | 12.87        | 25.50 (−7.93 ↓)           |
>
>
> The results show that a fixed tolerance of 0.6 is overly restrictive, degrading robustness, while 1.5 is too loose, making LORE behave like FARE. Although fixed values can be tuned per setting, finding a universal optimum is difficult. In contrast, the sample-specific margin ($\text{LORE}^2$+$m(x)$, $\text{LORE}^4$+$m(x)$, $\rho=0.1$) adapts to embedding scale, consistently improving robustness under various perturbations while maintaining clean accuracy and offering greater stability across training $\epsilon$ values and model types.
>
> # Response to the Strengths And Weaknesses:
>
> We appreciate your feedback on Section 4 effectively motivating our method and your recognition of its ability to achieve robust results with minimal clean accuracy degradation across diverse settings.
>
> > **In the experiment part, the authors use the image classification task as the measurement to verify the robustness. Since the CLIP representation has been used in various applications such as detection and VLM, could the authors validate the robustness on those tasks?**
>
> This is an insightful point that has also been raised by another reviewer, and we fully agree on its importance. While our initial evaluation focused on the foundational aspect of visual encoder robustness through image classification, zero-shot capabilities, and out-of-distribution generalization (relevant to VLM applications), we acknowledge that the utility of models like DINOv2 and CLIP extends to various downstream tasks, such as segmentation.
>
> During the rebuttal period, we conducted additional experiments to assess LORE's effectiveness on these more diverse downstream tasks for robust models. We found that LORE-hardened models maintain their strong performance even when evaluated on tasks beyond classification. However, due to time constraints, these evaluations are not as comprehensive as our previous experiments and focused specifically on the ViT-S/14 model, which was fine-tuned for 2 epochs on ImageNet for these new task-specific assessments. The results of these evaluations are presented in the table below, further demonstrating the generalizability and practical utility of LORE. This required dedicated experimental setups for each task, involving specialized datasets and metrics, which we are now able to include. The table shows the results of DINOv2 ViT-S/14 on the ADE20k dataset.
>
> | Method | Clean mIoU | EmbedAttack mIoU |
> |---|---|---|
> | Pretrained| 0.42 | 0.01 |
> | $\text{LORE}^8$|  0.23 | 0.15 |
> | $\text{FARE}^8$|  0.15 | 0.11 |
>
>
> > **There are many other related methods proposed to improve the robustness of visual encoders [1,2], so I suggest you compare your method with more baselines rather than only comparing with FARE.**
>
> We appreciate your suggestion. Reference [1] is the original paper introducing the FARE method, which we have used as our primary benchmark and discussed in detail throughout our paper. Regarding [2], unfortunately, we could not include it in our comparison as the paper is not publicly available (there isn't on arXiv and also it isn't accessible through other academic sources).
>
> It is also worth noting that TeCoA [3] was compared directly in the FARE paper, where FARE demonstrated competitive or superior performance over TeCoA. Since our method, LORE, consistently outperforms FARE across multiple metrics and setups, this indirectly highlights its strength relative to TeCoA as well. Due to space constraints, we did not replicate TeCoA in our paper, but we acknowledge it as a relevant supervised baseline.
>
> [3] Mao, C., Geng, S., Yang, J., Wang, X. E., and Vondrick, C. Understanding zero-shot adversarial robustness for large-scale models. In ICLR, 2023.
>
> > **The novelty of this work is limited. There has already been the unconstrained version of the adversarial optimization problem. And the difference between the proposed loss function Eq. (6) and the regularized loss function Eq. (3) only lies in the trainable weight $\lambda(x)$**
>
> Thank you for the comment. We respectfully disagree with the notion that the novelty of our work is limited. Our approach addresses a critical limitation in standard unsupervised adversarial fine-tuning frameworks, particularly in how they handle the trade-off between robustness and nominal (clean) performance.
>
> In the FARE method, the regularization coefficient $\lambda$ is effectively set to 0, meaning there is no mechanism to explicitly control the trade-off between clean and adversarial performance, and no such regularization schemes are introduced in the paper. While Eq. (3) introduces a constant $\lambda$ as a regularizer, we show in Section 4 that using a fixed $\lambda$ leads to suboptimal performance and instability.
>
> In contrast, our method proposes a sample-dependent and adaptive $\lambda$, modeled as a trainable dual network (Eq. 6), which offers several key advantages:
>
> - Formulating the problem as a constrained optimization problem, not just an ad hoc regularization of the original objective.
> - It provides principled control over the robustness–accuracy trade-off via constrained optimization (Section 6.1).
> - Stabilizing the network during training, whenever needed throughout training, only activated when the model is losing its nominal performance (Figure 10, 11. Appendix H)
> - In Figure 1b, we empirically show that by adaptively adjusting $\lambda$ (the dual function), LORE achieves **a strictly better Pareto frontier** than naively regularized losses with **any** fixed $\lambda$
> - As demonstrated in Section G.1, sample-based regularization performs significantly better than input-independent formulations.
> - It improves Out-of-Distribution Robustness, Embedding Interpretability, Image Classification accuracy and by dynamically adjusting the regularization strength based on each input sample (Section 6.2, 6.3).
>
> Despite the structural simplicity of our approach, we emphasize that LORE achieves strong empirical performance across multiple backbones and datasets, as shown in Section 6 and Appendix J.
>
> Additionally, we provide theoretical justification for LORE in:
>
> - Section 6.1, which presents a formal analysis of the controllability of the trade-off via the constraint threshold $\rho$, and
> - Appendix B, where we analyze the impact of dual variable parameterization on suboptimality bounds.
>
> Overall, we believe our formulation offers a practical and theoretically grounded improvement over previous methods, resolving a previously overlooked issue in unsupervised adversarial fine-tuning.
>
> > **In Line 130, the embedding space is denoted as ‘$\mathcal{Y}$’, while ‘$y$’ is used to represent the label of a data point ‘$x$’. It would be better to distinguish these two.**
>
> Thanks for the suggestion. We will address it in the final version.

---

> > ### Comment · Reviewer_TnAL · 2025-08-04
> >
> > I thank the authors for their detailed response and the additional experiments. My main concerns regarding the novelty and effectiveness on other tasks have been addressed. However, I remain curious about three additional questions:
> >
> > * Your formulation of Eq. (5) for the unsupervised setting is an extension of the classification setting in Eq. (4). Could the constrained problem in Eq. (4) also be solved using a Lagrangian optimization framework, similar to the dual optimization approach in Eq. (6)? Or has this already been explored in prior work?
> >
> > * In your method, you focus on optimizing the vision encoder. I am wondering whether it would also be possible to optimize both the vision and language encoders of CLIP to achieve better alignment and improved multi-modal robustness after applying the method. Do you have any initial ideas or suggestions regarding this? (No additional experiments are needed for this question.)
> >
> > * Additionally, in Lines 242–248, you claim that the average cosine similarity between clean image embeddings and their corresponding text embeddings is improved by LORE. Does this imply that the clean accuracy of LORE is even higher than that of the **original CLIP**? If so, what is the intuitive reason behind this improvement?

---

> > > ### Author Response · Authors · 2025-08-04
> > >
> > > We sincerely thank the reviewer for carefully reading our rebuttal and raising these thoughtful follow-up questions.
> > >
> > > > Answer for Q1
> > >
> > > Great question! Prior works have explored Lagrangian optimization in related settings, but typically with a varying scalar as the dual variable. Our contribution is to extend this idea by introducing a sample-dependent, adaptive dual function (specific architecture) that automatically adjusts the constraint strength. As we detail in “Answer for C1” to Reviewer CYtr, this formulation is not limited to the unsupervised case in Eq. (5). We have also applied LORE’s dual optimization framework to the supervised classification setting (Eq. 4) using TeCoA [1] as the base method. This adaptation significantly improved both robustness and nominal performance, showing that LORE is applicable to both unsupervised and supervised adversarial fine-tuning. For full experimental details and results, please refer to that section.
> > >
> > > > Answer for Q2
> > >
> > > Thank you for the insightful question. We have discussed a very similar point with Reviewer CYtr, and our thoughts there are largely applicable here. In summary, while our current work focuses on improving robustness on the vision side, LORE could in principle be extended to jointly fine-tune both the image and text encoders. However, prior findings (e.g., TeCoA [1], FARE [2]) show that simultaneous updates can cause training instability and drift in cross-modal alignment, especially in zero-shot settings.
> > >
> > > That said, in supervised scenarios—where text labels are available—joint fine-tuning becomes more feasible. In such cases, LORE’s proximity-based regularization could help stabilize updates to both encoders, keeping them anchored to their pretrained distributions while allowing controlled adaptation. For the full discussion, please refer to “Answer for Q1” in our response to Reviewer CYtr.
> > >
> > > > Answer for Q3
> > >
> > > In this experiment, we observed that adversarial fine-tuning—whether with FARE or LORE—tends to improve the embedding interpretability of the visual encoder. To assess embedding interpretability and cross-modal alignment, we used multiple natural-language templates to represent each class semantic. This is important because relying on a single fixed template (e.g., "This is a photo of {}") would limit the text encoder’s ability to represent class semantics accurately in the embedding space.
> > >
> > > However, this finding should not be interpreted as implying that fine-tuning with LORE (or FARE) necessarily improves clean classification accuracy. As shown in Tables 3, 9, and 10, clean accuracy may drop depending on the setting. The primary takeaway is the enhancement of semantic alignment, rather than a guaranteed gain in clean accuracy.
> > >
> > > Interestingly, this result was surprising to us because LORE does not use any labels or external information—it relies solely on the pretrained (anchored) encoder. Yet, the model optimized through this procedure produces embeddings with better interpretability than its teacher (anchored model).
> > >
> > >
> > >
> > > -----------------
> > > [1] Mao, C., Geng, S., Yang, J., Wang, X. E., and Vondrick, C. Understanding zero-shot adversarial robustness for large-scale models. In ICLR, 2023.
> > >
> > > [2] Christian Schlarmann, Naman Deep Singh, Francesco Croce, and Matthias Hein. Robust clip: Unsupervised adversarial fine-tuning of vision embeddings for robust large vision-language models. ICML, 2024.

---

> > > > ### Comment · Reviewer_TnAL · 2025-08-05
> > > > **Thanks for the response**
> > > >
> > > > Thank you for your response. Regarding Q3, I am also surprised by the improvement in interpretation, given that no text labels are used during the fine-tuning stage.
> > > >
> > > > I am still a bit confused about why the cosine similarity between the embeddings of clean images and the text prompts does not directly correspond to the clean accuracy defined in your paper. Could you clarify the detailed difference?

---

> > > > > ### Author Response · Authors · 2025-08-05
> > > > >
> > > > > Thank you for this follow‑up comment. You are indeed correct, this observation is also surprising to us, as this experiment may reveal improved semantic alignment even if top‑1 classification accuracy (based on a fixed prompt) remains the same or slightly decreases. We appreciate your interest in understanding it more deeply.
> > > > >
> > > > > While cosine similarity between clean image embeddings and text prompts reflects the absolute alignment in the joint embedding space, clean accuracy depends on the relative ranking of similarities across all class prompts. Thus, improvements in average cosine similarity (with correct class) indicate that, on average, the correct image–text pairs are drawn closer in the embedding space, but this does not necessarily guarantee that the cosine similarity between a clean image and text embeddings of incorrect classes will not also increase, or that the correct class will always have the highest similarity.
> > > > >
> > > > > Furthermore, recent work, including the follow-up on the FARE paper titled "Adversarially Robust CLIP Models Can Induce Better (Robust) Perceptual Metrics," [3] shows that adversarial training of CLIP models leads to perceptual metrics that are both more robust and more interpretable.
> > > > > A significant finding is that unsupervised adversarial fine tuning methods like FARE enhance interpretability. This is demonstrated through feature inversion, where a robust model, can produce semantically correct image reconstructions from an embedding. In contrast, the original, non-robust CLIP model yields only adversarial noise. The authors suggest that this is because adversarial training forces the model to emphasise robust, semantically meaningful features while suppressing non-robust features that lack semantic information. This also challenges the common assumption of a trade-off between clean and robust performance, as adversarial training is shown to improve both in this context.
> > > > >
> > > > > Overall, we see this as an interesting phenomenon worth investigating further, as it suggests that adversarial fine‑tuning with LORE may strengthen the semantic coherence of embeddings beyond what is captured by classification accuracy alone.
> > > > >
> > > > > -----
> > > > > [3]  Francesco Croce, Christian Schlarmann, Naman Deep Singh, and Matthias Hein. Adversarially robust clip models can induce better (robust) perceptual metrics. In SaTML, 2025.

---

> > > > > > ### Comment · Reviewer_TnAL · 2025-08-05
> > > > > >
> > > > > > Thank you. Your explanation makes sense to me. I suppose one possible reason for the improved alignment is that the embeddings of all images become closer after fine-tuning. Since adversarial training produces smoother representations, the distances between different image embeddings decrease, which may lead to the phenomenon you observed. This is an interesting observation, and although we do not yet have a clear explanation for it, I suggest including a more detailed discussion of this point in the revision.

---

> > > > > > > ### Author Response · Authors · 2025-08-05
> > > > > > >
> > > > > > > Thank you for providing this helpful intuition and for your suggestion. We agree that your interpretation is a plausible explanation for the observed phenomenon. We will include a more detailed discussion of this point in the final revision. Please feel free to let us know if anything remains unclear or if further details would be helpful.

---

### Official Review · Reviewer_CYtr · 2025-07-02

**Clarity:** 3
**Significance:** 4
**Originality:** 3
**Rating:** 5
**Confidence:** 5

**Summary:**

The paper proposes a novel unsupervised adversarial fine-tuning framework dubbed Lagrangian-Optimised Robust Embeddings (LORE). The authors aim to address the limitations of current unsupervised adversarial fine-tuning approaches: instability during early stages of fine-tuning and suboptimal trade-off between clear and adversarial performance. LORE maintains the performance on clean data during the fine-tuning process by enforcing embedding proximity constraints.  Moreover, LORE enables principled control over the clean-adversarial performance trade-off. The paper demonstrates the effectiveness of this approach on CLIP and DinoV2 models, compared with the SOTA FARE approach.

**Questions:**

Q1: Can LORE be extended to simultaneously fine-tune both image and text encoders in Visual Language Models like CLIP?

Q2:  "(2) LORE relies on a high-quality pretrained model, and its performance, especially on perceptual metrics, is sensitive to the fidelity of that model." Can you please provide more discussion on that matter? At what point does LORE stop being an effective approach?

**Ethical Concerns:**

["NO or VERY MINOR ethics concerns only"]

**Final Justification:**

**I clearly recommend accepting the paper.** LORE tackles an important challenge in adversarial fine-tuning: maintaining clean accuracy during the early stages of training. The authors submitted a strong paper and followed up with a rebuttal that effectively addressed my concerns. This includes promising results in supervised settings and on downstream tasks beyond classification, which also responds to concerns raised by other reviewers. In my view, the paper has clear practical value. For example, I see potential for LORE in applications such as model merging, where extended fine-tuning can compromise the integrity of merged models. LORE could enable the creation of robust task vectors while preserving clean data performance due to addressing the performance drop in the early stages of training.

**Overall, I clearly recommend accepting the paper and kindly ask other Reviewers to consider increasing their scores.**

**Limitations:**

The authors adequately addressed the limitations. However, I feel that more discussion on the following would improve the paper: " (2) LORE relies on a high-quality pretrained model, and its performance, especially on perceptual metrics, is sensitive to the fidelity of that model."

**Paper Formatting Concerns:**

No concerns.

**Quality:**

4

**Strengths And Weaknesses:**

### Strengths

S1: LORE effectively addresses a crucial problem of instability during the early stages of adversarial fine-tuning in unsupervised adversarial fine-tuning. I find the significance of this result to be high, as it enables adversarial fine-tuning for a limited number of updates without collapsing the model’s performance. This makes the adversarial fine-tuning process more practical and user-friendly.

S2: LORE effectively addresses an important problem of suboptimal trade-off between clear and adversarial performance, pushing the empirical Pareto front.

S3: Strong experimental section. (However, see Weaknesses for ideas of additional improvements)

### Weaknesses

W1: Optimal trade-off claim.  The paper uses strong language to describe the performance of LORE. For example, “LORE yields a strictly better Pareto front, achieving optimal trade-offs between robustness and clean accuracy” (Figure 1). At the same time, the paper notes that: "(1) A deeper theoretical analysis of the trade-offs in unsupervised adversarial finetuning is needed" (L294-295).  While the empirical results indeed prove that the Pareto front is improved over the baseline, calling it “optimal” might be an overstatement. According to my understanding, there is no proof that LORE finds the global optimum of the robustness vs. accuracy trade-off.

W2: Missing broader related work. I believe that the Related Works section misses some earlier attempts at ensuring adversarial robustness in the unsupervised fine-tuning setting. For example:

[1] Kim et al., Adversarial Self-Supervised Contrastive Learning, NeurIPS 2020

[2] Zhang et al., Adversarial Contrastive Learning for Unsupervised Representation Learning, NeurIPS 2021

[3] Chiang et al., Robust Contrastive Learning for Unsupervised Visual Representation Learning, CVPR 2022


W3: Limited downstream tasks. Dinov2 can produce an embedding usable in a variety of downstream tasks, e.g segmentation. Yet, the evaluation in the paper is limited to classification. I believe it would be valuable to asses the effectivness of robust models (e.g. DinoV2) obtained with LORE on more diverse downstream tasks, e.g. similarly to [4].

[4] Kowalczuk et al., Benchmarking Robust Self‑Supervised Learning Across Diverse Downstream Tasks, CVPR 2024 Workshop on Robust Representation Learning

### Comments
C1: While this is not a weakness per se, LORE targets instability during early stages of adversarial fine-tuning and suboptimal trade-off between clear and adversarial performance in the *unsupervised* adversarial fine-tuning. According to my understanding, this approach is not directly applicable to supervised models. Yet, the supervised setting also suffers from similar problems. This limits the (still high) significance of the method proposed in this paper. The paper's significance would be even greater if a LORE-like approach was applicable to all adversarial fine-tuning scenarios.

---

> ### Author Rebuttal · Authors · 2025-07-31
>
> Thank you for your careful reading of our paper and for highlighting both its strengths and areas for improvement. We respond below to your main concerns and detail how we will strengthen the manuscript.
>
> > **Answer for W1**
>
> We agree with your observation and have revised the caption of Figure 2 to remove the term “optimal,” which could misleadingly imply a proven global optimum that we do not formally establish. As shown in our experiments (Section 6), our goal is to present an approach that consistently outperforms the FARE baseline in practice, emphasizing empirical improvements without overstating theoretical guarantees.
>
> > **Answer for W2**
>
> We sincerely appreciate your suggested references. The first paper you mentioned (Kim et al., NeurIPS 2020) is indeed relevant, and we will include it in the extended version of our Related Work section. However, we were unable to locate the last two references as cited. If you could clarify or provide more details, we would be happy to consider them as well.
>
> > **Answer for W3**
>
> The authors acknowledge the importance of evaluating LORE's effectiveness on diverse downstream tasks beyond image classification, such as segmentation, for models like DINOv2. During the rebuttal, they conducted additional, though time-constrained and less comprehensive, experiments focusing on the ViT-S/14 model fine-tuned for 2 epochs on ImageNet. These new evaluations show that LORE-hardened models maintain strong performance on these tasks, demonstrating the general utility of LORE. The table shows the results of DINOv2 ViT-S/14 on the ADE20k dataset. We used EmbedAttack to evaluate robustness, following the approach in [4].
>
> | Method | Clean mIoU | EmbedAttack mIoU |
> |---|---|---|
> | Pretrained| 0.42 | 0.01 |
> | $\text{LORE}^8$|  0.23 | 0.15 |
> | $\text{FARE}^8$|  0.15 | 0.11 |
>
>
> > **Answer for C1**
>
> We thank the reviewer for this insightful comment and fully agree that extending LORE to supervised adversarial fine-tuning would broaden its impact. While LORE was originally developed in an unsupervised setting, its core principle is potentially compatible with supervised adversarial training.
>
> ### **Experimental Extension:**
>
> ### **1. LORE in Supervised Adversarial Fine-Tuning**
>
> To directly address the reviewer’s suggestion, we extended our work by applying LORE regularization to the supervised adversarial fine-tuning method TeCoA [1], which aligns image-text pairs using supervised losses.
>
> Given an image-label pair $(x, y)$, TeCoA optimizes a contrastive loss to align adversarial image embeddings with text embeddings. The loss function is defined as:
>
> \begin{equation}
> L_{sup}(x, y; \theta) = \max_{\delta \in \Delta} - \sum_i \Bigg[y_i \log \frac{\exp(\cos(z, \psi(t_i))/\tau)}{\sum_k \exp(\cos(z, \psi(t_k))/\tau)} \Bigg],
> \end{equation}
>
> where $z = \phi_\theta(x + \delta)$ and $\psi(t_i)$ are the image and text embeddings, respectively. Here, $\psi(t_i)$ corresponds to the text embedding of the *i-th* class in the dataset, where each class label is inserted into a fixed natural language template, e.g., *"This is a photo of {class}"*.
>
> ---
>
> We incorporated our proximity-based regularizer into TeCoA, forming a combined method *(TeCoA + $l_2$)* as follows:
>
> $$
> \max_{\omega\in\Omega}\min_{\theta\in\Theta} E_{x,y \sim D} \left[ L_{\text{sup}}(x,y;\theta) + \lambda_\omega(x) \big(\|\phi_\theta(x)- \phi_0(x)\|_2^2 - \rho  \|\phi_0 (x)\|_2^2   \big) \right]
> $$
>
> ### **2. Distribution-Level Constraint**
>
> We also explored a probability-distribution variant of LORE for the supervised setting, constraining the KL divergence between the fine-tuned model’s output distributions and those of the pretrained model. This approach requires only class text embeddings, not sample labels. Applied to TeCoA, it yields a second regularized variant: TeCoA + KL.
>
> For classification using probability distributions induced by the image classifier or a contrastive model (outputs logits of softmax of cosine similarities), for both current and original models, We formulate the constrained optimization problem:
>
> $$
> \min_{\theta \in \Theta} \;
> E_{x,y \sim D}\big[L_{sup}(x,y;\theta)\big] \quad \text{s.t.} \quad
> D_{\text{KL}}(p_{\theta_0}(\cdot \mid x)\,\|\,p_\theta(\cdot \mid x)) \leq \rho\cdot m(x) , \quad \text{where} \quad m(x) = H(p_{\theta_0}(\cdot \mid x))
> $$
>
> And building on this formulation, LORE's final objective function can be computed as before. The framework adjusts the proximity constraint by model confidence: low entropy (high confidence) tightens the constraint, while high entropy relaxes it for greater adaptation flexibility.
>
> ## Results
>
> **Experimental setting:** We fine-tuned ViT-B/32 CLIP models for 15 epochs on ImageNet-100 using a learning rate of 1e-5 and perturbation strength ε. The proximity constraint hyperparameter used ρ=0.1, consistent with previous LORE experiments.
>
> | Training ε | Method                 | Clean                  | ε=1   | ε=2   | ε=4   | ε=8   | Avg Robust Acc           |
> |------------|-----------------------|-----------------------|-------|-------|-------|-------|--------------------------|
> | 2          | **TeCoA**              | 60.04 (ref)                | 49.11 | 35.94 | 14.73 | 0.50  | 25.07    (ref)               |
> | 2          | **TeCoA + $l_2$**       | 75.97 (**+15.93 ↑**)  | 53.24 | 39.12 | 18.12 | 1.23  | 27.93 (**+2.86 ↑**)     |
> | 2          | **TeCoA + KL**     | 66.50 (**+6.46 ↑**)   | 53.04 | 39.53 | 16.94 | 0.60  | 27.53 (**+2.46 ↑**)     |
> | 4          | **TeCoA**             | 49.55  (ref)               | 44.20 | 38.21 | 19.80 | 1.03  | 25.81        (ref)           |
> | 4          | **TeCoA + $l_2$**       | 73.12 (**+23.57 ↑**)  | 56.20 | 53.12 | 49.12 | 5.23  | 40.92 (**+15.11 ↑**)    |
> | 4          | **TeCoA + KL**     | 56.01 (**+6.46 ↑**)   | 48.62 | 42.03 | 22.18 | 1.25  | 28.52 (**+2.71 ↑**)     |
> | 8          | **TeCoA**              | 30.22         (ref)        | 24.72 | 19.09 | 7.41  | 2.35  | 13.39         (ref)          |
> | 8          | **TeCoA + $l_2$**       | 67.23 (**+37.01 ↑**)  | 36.21 | 26.40 | 15.12 | 4.23  | 20.49 (**+7.10 ↑**)     |
> | 8          | **TeCoA + KL**  | 38.06 (**+7.84 ↑**)   | 27.69 | 21.94 | 8.75  | 2.82  | 15.30 (**+1.91 ↑**)     |
>
> As shown above, TeCoA consistently benefits from both LORE and KL regularization, achieving improved robustness while maintaining clean performance. These results demonstrate that LORE is not limited to unsupervised fine-tuning and that proximity-based constraints—whether applied in embedding space or probability space—can effectively stabilize adversarial fine-tuning in supervised frameworks as well.
>
> > **Answer for Q1**
>
> Our primary focus in this work was improving robustness on the vision side, but your suggestion opens an important direction: extending LORE to jointly fine-tune both image and text encoders. Prior findings (e.g., TeCoA [1]) indicate that such joint tuning can cause training instability, especially in zero-shot settings where cross-modal alignment may drift and generalization degrades. Similarly, FARE [2] emphasizes the need to preserve the original embedding characteristics of both clean and adversarial examples in VLM pipelines (e.g., CLIP in LLaVA). Simultaneous fine-tuning risks keeping modalities aligned with each other while drifting away from their pretrained distributions, harming robustness and semantic fidelity. Nonetheless, it could offer benefits in scenarios such as image-only attacks, where updating the text encoder may help realign embeddings with perturbed image features, potentially mitigating attack effects.
>
> ## On Extending LORE:
>
> **In Unsupervised Settings:**
>
> Extending LORE to jointly fine-tune both encoders in a fully unsupervised setting is non-trivial. Without access to text supervision (e.g., class labels or prompts), it's unclear how to meaningfully update the text encoder. In this case, LORE would not have the necessary guidance to preserve alignment or constrain both encoders effectively.
>
> **In Supervised Settings:**
>
> In contrast, supervised settings offer a viable path forward. Since textual labels and their embeddings are available, the text encoder can be updated effectively. For instance, in our proposed extensions to TeCoA (discussed in response to Comment C1), applying LORE’s proximity-based regularization can help stabilize joint fine-tuning. The constraints act as stabilizers, keeping both encoders anchored to their pretrained representations while allowing controlled adaptation.
>
> Moreover, if the threat model includes perturbations to both modalities, then fine-tuning the text encoder becomes necessary, and LORE-style regularization could be critical for maintaining dual robustness.
>
> > **Answer for Q2**
>
> Both LORE and FARE's effectiveness hinges on the quality of their frozen pretrained reference model, which acts as an anchor. LORE will still converge and improve robustness, as it helps control the nominal-adversarial robustness trade-off by constraining model deviation; however, its performance ceiling (both clean accuracy and robustness) is inherently limited by the reference model's quality. There's no sharp cutoff; LORE's benefits gradually lessen with lower-quality pretrained models. Empirical evidence from Tables 1 and 2 consistently supports this: stronger pretrained models (e.g., higher clean accuracy, larger architectures like ViT-B/16 over ViT-B/32, or DINOv2 ViT-B/14 over ViT-S/14) consistently lead to better nominal and robust results when LORE is applied.
>
> ---
> [1] Mao, C., Geng, S., Yang, J., Wang, X. E., and Vondrick, C. Understanding zero-shot adversarial robustness for large-scale models. In ICLR, 2023.
>
> [2] Christian Schlarmann, Naman Deep Singh, Francesco Croce, and Matthias Hein. Robust clip: Unsupervised adversarial fine-tuning of vision embeddings for robust large vision-language models. ICML, 2024.
>
> [4] Kowalczuk et al., Benchmarking Robust Self‑Supervised Learning Across Diverse Downstream Tasks, CVPR 2024

---

> ### Comment · Reviewer_CYtr · 2025-08-02
>
> Thank you for your answer.
>
> ### References
> I apologize for the earlier incorrect references. Below are the correct ones. My intention was to bring to the authors’ attention a relevant, though somewhat parallel, line of work [1, 2, 3, 4]. I believe it offers useful background for robust self-supervised learning, which is increasingly important given the large volume of work currently being published in deep learning.
>
> [2] Zhang et al., *Decoupled Adversarial Contrastive Learning for Self-supervised Adversarial Robustness*, ECCV 2022
> [3] Kim et al., *Adversarial Self-Supervised Contrastive Learning*, NeurIPS 2020
>
> ### Other Tasks
> This concern was also raised by reviewer TnAL28. I find the authors’ response reasonable given the limited time during the rebuttal. I would kindly suggest expanding the comparison to include additional tasks, such as those in [4] and [5], as this would significantly strengthen the paper.
>
> [4] Kowalczuk et al., *Benchmarking Robust Self‑Supervised Learning Across Diverse Downstream Tasks*, CVPR 2024
>
> [5] Oquab et al., *DINOv2: Learning Robust Visual Features without Supervision*, 2024
>
>
> ### On Extending LORE
> I appreciate the additional discussion and experiments on extending LORE. This clearly strengthens the method in my view.
>
>
> ### Summary
> Overall, I would like to see the paper accepted. While I am not comfortable with increasing my already strong score to the maximum 6/6, I will actively participate in the discussion and advocate for other Reviewers to consider raising their ratings.
>
> I find that addressing the performance drop on clean data during the early stages of adversarial fine-tuning is an important contribution with practical implications. For instance, it could be applied in the context of Model Merging, where pretrained self-supervised models can be fine-tuned using LORE for just a few steps—as required by Model Merging—to improve robustness without compromising clean accuracy. I strongly suggest that the authors make their code as user-friendly as possible and provide trained checkpoints to increase the impact of their work, given acceptance.

---

### Comment · Reviewer_CYtr · 2025-08-03

Dear Reviewers and AC

*As a Reviewer, I clearly recommend accepting the paper. I kindly ask other Reviewers to consider increasing their scores.*

LORE tackles an important challenge in adversarial fine-tuning: maintaining clean accuracy during the early stages of training. In my opinion, the authors submitted a strong paper and followed up with a rebuttal that effectively addressed my concerns. This includes 1) promising results in supervised settings and 2) on downstream tasks beyond classification, which also responds to concerns raised by other Reviewers. In my view, the paper has clear practical value.

For example, LORE shows promise in Model Merging settings, where models are fine-tuned on different tasks and subsequently combined. In such cases, extended adversarial fine-tuning can cause models to diverge significantly from the base model, making effective merging difficult or even infeasible. On the other hand, limiting adversarial fine-tuning to preserve mergeability often leads to a sharp drop in clean accuracy during the initial adversarial training steps, which also degrades the quality of the merged model. By mitigating this early drop in clean performance, LORE makes short adversarial fine-tuning more practical and helps maintain the compatibility needed for successful merging.

---

### Author Response · Authors · 2025-08-09
**Summary of Rebuttal**

We sincerely thank all reviewers for their thorough engagement and constructive feedback on our submission, "LORE: Lagrangian-Optimized Robust Embeddings for Visual Encoders." We believe their insights have significantly helped us strengthen the manuscript.

Our paper introduces LORE, a novel unsupervised adversarial fine-tuning framework that addresses two critical limitations of existing approaches: instability during early training and suboptimal trade-offs between clean and adversarial performance. LORE achieves this through a principled constrained optimization framework using a dynamic, sample-dependent dual network, which offers fine-grained control over embedding-space proximity to a reference model. We showed that our method can achieve a better Pareto front than ad hoc regularization schemes.

It is important to emphasize that we have not altered our proposed method or any of its core claims during the rebuttal period. Instead, our efforts during the rebuttal period focused on clarifying existing aspects of our approach and expanding our empirical evaluations with new experiments to further support and validate our original claims.

Here are the key improvements and clarifications, conducted based on reviewers' suggestions:

* LORE Extension to Supervised Settings: We extended LORE's core principle to the supervised adversarial fine-tuning setting, applying it to the TeCoA [1] framework. New experimental results show that proximity-based regularization significantly improves both robustness and clean performance in a supervised context, demonstrating LORE's broader applicability. (Addressing Reviewer CYtr)
* Refined Discussion and Contextualization: We've clarified existing claims (e.g., rephrasing the "optimal" Pareto front to emphasize significant empirical improvement), provided deeper discussions on specific experimental observations (like '0' results in Table 3, the nuanced relationship between improved cosine similarity and classification accuracy, and the impact of the pretrained model's quality on LORE's effectiveness), and will expand the Related Work section to include additional relevant earlier attempts at unsupervised adversarial robustness, ensuring a more comprehensive overview. (Addressing Reviewer TnAL, Reviewer CYtr, and Reviewer Jf2T)
* Expanded Evaluation on Downstream Tasks: To address requests for broader evaluation beyond classification, we performed new experiments on a segmentation task (ADE20k dataset) using a LORE-hardened DINOv2 ViT-S/14 model. These results further demonstrate LORE's ability to maintain strong performance in diverse downstream applications, showcasing its general utility. While time-constrained, these new results confirm LORE's transferability. (Addressing Reviewer TnAL, and  Reviewer CYtr)
* Enhanced Clarity on Novelty: We provided further explanation on the distinct novelty of LORE's sample-dependent, adaptive dual network ($\lambda_{\omega}(x)$), highlighting how it fundamentally differs from heuristic fixed-weight regularization and is crucial for LORE's stability and superior trade-offs, with further experiments justifying our choice of a sample specific margin. (Addressing Reviewer TnAL)

We are confident that these clarifications and the inclusion of new experimental results have improved the paper's clarity, comprehensiveness, and overall quality, thereby reinforcing the validity of our original claims. LORE is believed to constitute a significant and practically relevant contribution to the discipline of robust machine learning.
Thank you again for your time and consideration.

Sincerely,

The Authors

---

[1] Mao, C., Geng, S., Yang, J., Wang, X. E., and Vondrick, C. Understanding zero-shot adversarial robustness for large-scale models. In ICLR, 2023.

---

### Decision · Program_Chairs · 2025-09-17

**Decision:**

Accept (poster)

**Comment:**

This paper proposes LORE, a principled adversarial fine-tuning framework using sample-dependent constrained optimization. It addresses early instability and robustness–accuracy trade-offs with strong empirical results across classification, segmentation, and supervised settings.
Reviewers raised major concerns around novelty, downstream evaluation, and baseline coverage.
Authors added segmentation results, supervised extensions (TeCoA+LORE), and clarified theoretical and empirical distinctions from prior work like FARE.
Given the practical use-case and clarity of the paper, I recommend acceptance.